# Inference-Time Alignment Control for Diffusion Models with Reinforcement Learning Guidance

## Abstract

Denoising-based generative models, particularly diffusion and flow matching algorithms, have achieved remarkable success. However, aligning their output distributions with complex downstream objectives remains challenging. While reinforcement learning (RL) fine-tuning methods, inspired by advances in RL from human feedback (RLHF) for large language models, have been adapted to these generative frameworks, current RL approaches offer limited flexibility in controlling alignment strength after fine-tuning. In this work, we view RL fine-tuning for diffusion models through the lens of stochastic differential equations and implicit reward conditioning. We introduce *Reinforcement Learning Guidance* (RLG), an inference-time method that adapts Classifier-Free Guidance (CFG) by combining the outputs of the base and RL fine-tuned models via a geometric average. Our theoretical analysis shows that RLG's guidance scale is mathematically equivalent to adjusting the KL-regularization coefficient in standard RL objectives, enabling dynamic control over the alignment-quality trade-off without further training. Extensive experiments demonstrate that RLG consistently improves the performance of RL fine-tuned models across various architectures, RL algorithms, and downstream tasks, including human preferences, compositional control, compressibility, and text rendering. Furthermore, RLG supports both interpolation and extrapolation, thereby offering unprecedented flexibility in controlling generative alignment. Our approach provides a practical and theoretically sound solution for enhancing and controlling diffusion model alignment at inference. The source code for RLG is available in the anonymous repository: [1].

## 1 Introduction

While denoising-based generative models—primarily diffusion Ho et al. (2020); Rombach et al. (2022) and flow matching Lipman et al. (2022); Esser et al. (2024) algorithms—have gained widespread usage, a key challenge is aligning their learned distribution with complex downstream objectives such as human preferences Kirstain et al. (2023), compositional correctness Ghosh et al. (2023), text rendering Liu et al. (2025b), or data compressibility Black et al. (2023). Existing approaches include reward-weighted regression Peng et al.; Lee et al. (2023); Fan et al. (2025), direct reward fine-tuning Xu et al. (2023); Prabhudesai et al. (2023); Clark et al. (2023), and reinforcement learning (RL) fine-tuning.

Owing to significant advancements in Reinforcement Learning from Human Feedback (RLHF) Black et al. (2023); Lee et al. (2023) for Large Language Models (LLMs), RL has been adapted to diffusion models by formulating denoising as a multi-step decision-making process, enabling algorithms like REINFORCE Williams (1992); Mohamed et al. (2020); Black et al. (2023), Direct Preference Optimization (DPO) Rafailov et al. (2023); Wallace et al. (2024), and Group Relative Policy Optimization (GRPO) Shao et al. (2024); Liu et al. (2025b)—to diffusion models. However, current RL methods for diffusion models still exhibit several limitations, primarily in two respects. First, the exact probability of a sampled image is intractable due to the nature of diffusion algorithms, which undermines the effectiveness of existing RL algorithms Black et al. (2023); Gong

---

[1]https://anonymous.4open.science/r/Reinforcement-learning-guidance-7B5A/

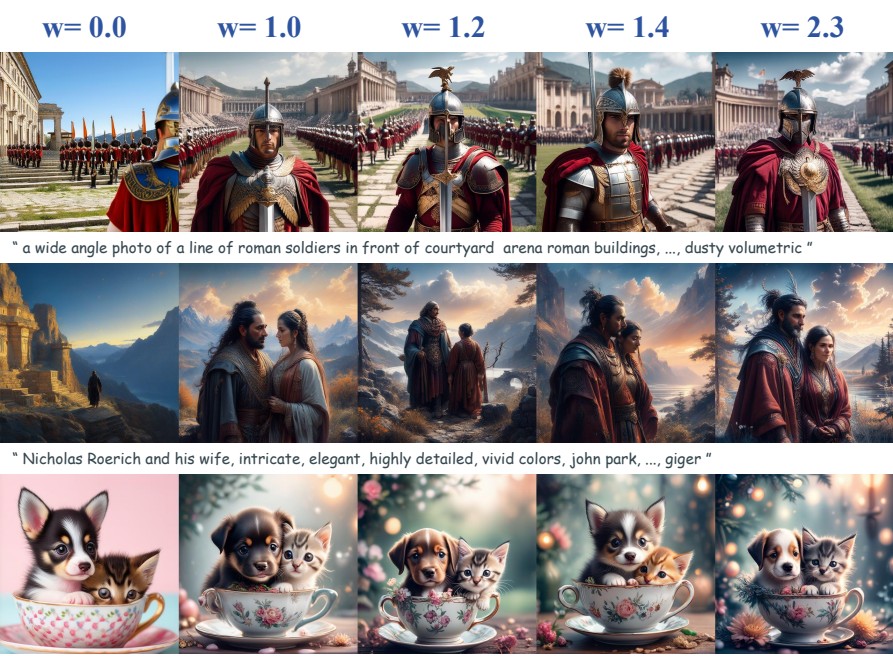

w= 0.0    w= 1.0    w= 1.2    w= 1.4    w= 2.3

" a wide angle photo of a line of roman soldiers in front of courtyard  arena roman buildings, ..., dusty volumetric "

" Nicholas Roerich and his wife, intricate, elegant, highly detailed, vivid colors, john park, ..., giger "

" a puppy and a kitten in a teacup together "

Figure 1: Selected qualitative results for the human preference alignment task using SD3.5-M with GRPO and our RLG. The PickScore is displayed on each image. As the RLG scale increases, the images generally become more detailed, aesthetically pleasing, which is corroborated by the rising PickScores.

et al. (2025). Second, the degree to which the base model aligns with downstream objectives remains fixed after RL fine-tuning and is sensitive to hyperparameter choices, such as the Kullback–Leibler (KL) coefficient. This inflexibility prevents users from dynamically balancing alignment and generation quality, which may be crucial in applications such as compressibility.

In this work, we draw inspiration from the stochastic differential equation (SDE) nature of denoising-based generative models Song et al. (2020b), which motivates us to interpret RL fine-tuning of diffusion models as a special case of generation conditioned on implicit rewards learned through reinforcement learning objectives Rafailov et al. (2024); Zhu et al. (2025); Cui et al. (2025). Building upon this perspective, we introduce an inference-time enhancement technique, *Reinforcement Learning Guidance* (RLG), which adapts the established controlling approach, Classifier-Free Guidance (CFG) Ho & Salimans (2022); Zheng et al. (2023), by computing a weighted geometric average of the outputs from the base model and the RL fine-tuned model. We theoretically demonstrate that this weighted averaging has the same effect as modifying the KL coefficient in RL fine-tuning, but crucially, it requires no additional training.

Empirical results on downstream tasks demonstrate that RLG enhances the performance of RL fine-tuned models across diverse tasks and setups: various model types (diffusion and flow matching), a range of RL methods (policy gradient, DPO, GRPO, etc.), and multiple downstream objectives (image aesthetics, compositional control, compressibility, text rendering, inpainting, and personalized generation). Furthermore, RLG supports both interpolation and extrapolation, thereby offering substantial flexibility in controlling the degree of alignment with downstream objectives.

Our contributions are summarized as follows:

- We propose *Reinforcement Learning Guidance* (RLG), a novel, training-free approach for enhancing and controlling the inference-time alignment of denoising-based generative models.
- We provide a theoretical foundation for RLG, demonstrating that its guidance scale is mathematically equivalent to adjusting the KL-regularization coefficient in the underlying RL objective. This analysis formally accounts for the effectiveness of extrapolation ($w > 1$).

- We perform extensive experiments on a diverse set of alignment tasks, showing that RLG consistently enhances performance by enabling models to surpass their original fine-tuned capabilities, while also allowing for flexible trade-offs between competing objectives.

## 2 BACKGROUND

### 2.1 DIFFUSION AND FLOW-BASED GENERATIVE MODELS

Diffusion Ho et al. (2020); Song et al. (2020a); Rombach et al. (2022) and flow-based Lipman et al. (2022); Liu et al. (2022); Esser et al. (2024) models generate data by transforming noise into samples via either stochastic (SDE) or deterministic (ODE) processes. Diffusion models, e.g., DDPM Ho et al. (2020), DDIM Song et al. (2020a), and Stable Diffusion Rombach et al. (2022), corrupt data with noise and train a network to learn the score function $\mathbf{s}(\mathbf{X}_t, t)$ for reverse denoising Song et al. (2020b). Flow-based approaches such as Flow Matching Lipman et al. (2022) learn a velocity field $\mathbf{v}(\mathbf{X}_t, t)$ to follow a deterministic path from prior to data Liu et al. (2022); Esser et al. (2024); Tong et al. (2023); Kong et al. (2024); Liu et al. (2025a); Wan et al. (2025).

A reference flow $(X_t)_{t \in [0,1]}$ interpolates between $X_1 \sim p_1$ and $X_0 \sim p_{\text{data}}$:

$$X_t = \beta_t X_1 + \alpha_t X_0, \quad \alpha_0 = \beta_1 = 0, \ \alpha_1 = \beta_0 = 1. \tag{1}$$

For ODE-based Flow Matching, the model is trained to match the reference velocity:

$$\mathbf{v}(\mathbf{X}_t, t) \approx \dot{\beta}_t X_0 + \dot{\alpha}_t X_1.$$

Diffusion models solve an SDE with noise schedule $\sigma(t)$, typically parameterized by $\alpha_t = \sqrt{\bar{\alpha}_t}$, $\beta_t = \sqrt{1 - \bar{\alpha}_t}$.

Formally, the velocity field in Flow Matching can be written in terms of the score function as

$$\mathbf{v}(\mathbf{x}, t) = \left( \frac{\dot{\alpha}_t}{\alpha_t} \right) \mathbf{x} + \beta_t \left( \frac{\dot{\alpha}_t}{\alpha_t} \beta_t - \dot{\beta}_t \right) \mathbf{s}(\mathbf{x}, t). \tag{2}$$

The two paradigms unify under the SDE Song et al. (2020b):

$$d\mathbf{X}_t = \left( \mathbf{v}(\mathbf{X}_t, t) - \frac{1}{2} \sigma(t)^2 \mathbf{s}(\mathbf{X}_t, t) \right) dt + \sigma(t) \, dw, \tag{3}$$

where $w$ is Brownian motion; diffusion and Flow Matching differ in $\mathbf{v}$, $\mathbf{s}$, and $\sigma(t)$.

### 2.2 GUIDANCE AND CONTROL IN GENERATIVE MODELS

Controlling generative model outputs is essential for conditional generation tasks. Early work such as Classifier Guidance (CG) steers the generation process using gradients from a separately trained classifier Dhariwal & Nichol (2021), but this approach is computationally costly and limited by the need for an external model.

Classifier-Free Guidance (CFG) has become the standard alternative Ho & Salimans (2022). At inference, CFG computes two passes: one with the actual condition $c$ (e.g., text-guided) and one with the null condition $\emptyset$. The guided velocity field $\hat{\mathbf{v}}_\theta$ is a linear interpolation between these two outputs:

$$\hat{\mathbf{v}}_\theta(\mathbf{x}_t, t | c) \triangleq (1 - \omega) \mathbf{v}_\theta(\mathbf{x}_t, t | \emptyset) + \omega \mathbf{v}_\theta(\mathbf{x}_t, t | c), \tag{4}$$

where $\omega$ is the guidance scale parameter. Setting $\omega = 1$ recovers conditional generation, while $\omega > 1$ extrapolates beyond the conditional prediction.

The same principle applies to the model's score function:

$$\hat{\mathbf{s}}_\theta(\mathbf{x}_t, t | c) \triangleq (1 - \omega) \mathbf{s}_\theta(\mathbf{x}_t, t | \emptyset) + \omega \mathbf{s}_\theta(\mathbf{x}_t, t | c), \tag{5}$$

where $\mathbf{s}_\theta(\mathbf{x}_t, t | c) = \nabla_{\mathbf{x}_t} \log p_\theta(\mathbf{x}_t | c)$. This can be equivalently written as:

$$\hat{\mathbf{s}}_\theta(\mathbf{x}_t, t | c) = \nabla_{\mathbf{x}_t} \log \left( p_\theta(\mathbf{x}_t)^{1-\omega} p_\theta(\mathbf{x}_t | c)^\omega \right), \tag{6}$$

Although CFG is highly effective, most existing methods focus on adherence to training-time conditions such as text condition, leaving open the possibility of leveraging reinforcement learning rewards as dynamic, flexible forms of guidance.

## 2.3 PREFERENCE ALIGNMENT IN GENERATIVE MODELS

Preference learning methods from LLMs have been adapted to fine-tune T2I diffusion models for human alignment. A pre-trained model $\pi_{\text{ref}}$ is fine-tuned to maximize a reward $R(\mathbf{x})$ under KL regularization:

$$\pi_\theta^* = \arg\max_{\pi_\theta} \mathbb{E}_{\mathbf{x}\sim\pi_\theta}[R(\mathbf{x})] - \beta\, D_{\text{KL}}(\pi_\theta\|\pi_{\text{ref}}), \tag{7}$$

where $\beta$ controls the reward–regularization trade-off.

The optimal solution to this problem shows the aligned policy is a re-weighted reference policy, with weights exponentially proportional to the reward Peng et al.; Lee et al. (2023); Fan et al. (2025):

$$p^*(\mathbf{x}) \propto p_{\text{ref}}(\mathbf{x})\exp\left(\frac{1}{\beta}R(\mathbf{x})\right). \tag{8}$$

This objective can be optimized with policy gradient methods include PPO Schulman et al. (2017); Black et al. (2023), as well as direct approaches such as DPO Rafailov et al. (2023); Wallace et al. (2024) and GRPO Sun et al. (2025); Shao et al. (2024). However, many alignment methods may overlook characteristics of diffusion models. For instance, Diffusion-DPO Wallace et al. (2024) is upper-bounded by the original DPO loss. In particular, integrating reinforcement learning objectives with diffusion-specific techniques—such as Classifier-Free Guidance (CFG) Ho & Salimans (2022)—remains underexplored, presenting opportunities to design approaches that combine reward-based alignment with the generative priors and guidance capabilities unique to diffusion.

**Concurrent Work.** While finalizing our paper, two concurrent works, CFGRL(Frans et al., 2025) and Diffusion Blend(Cheng et al., 2025), appeared. Both investigate inference-time manipulation techniques via score interpolation. However, CFGRL focuses solely on offline RL and simple task settings, while Diffusion Blend does not establish a connection between interpolation and implicit reward guidance. In contrast, RLG offers a comprehensive analysis of score interpolation from the perspective of implicit classifier guidance and demonstrates its effectiveness across various image generation models, RL algorithms, and tasks.

## 3 METHODS

### DERIVING REINFORCEMENT LEARNING GUIDANCE (RLG)

Let $r$ represent the desired attribute, such as a high preference score. Following Bayes' rule, the score function of the conditional distribution $p_{\text{ref}}(\mathbf{x}_t|r)$ can be decomposed as:

$$\nabla_{\mathbf{x}_t}\log p_{\text{ref}}(\mathbf{x}_t|r) = \nabla_{\mathbf{x}_t}\log p_{\text{ref}}(\mathbf{x}_t) + \nabla_{\mathbf{x}_t}\log p(r|\mathbf{x}_t). \tag{9}$$

To relate this to a reward function $R(\mathbf{x}_t)$, following Zhu et al. (2025), we model $p(r|\mathbf{x}_t)$ via an energy-based form:

$$p(r|\mathbf{x}_t) = \frac{\exp(R(\mathbf{x}_t))}{Z}, \quad Z = \int \exp(R(\mathbf{x}_t))\, d\mathbf{x}_t. \tag{10}$$

Substituting this into Equation 9 yields the general formula for reward gradient guidance:

$$\hat{\mathbf{s}}(\mathbf{x}_t, t) = \mathbf{s}_{\text{ref}}(\mathbf{x}_t, t) + \eta\nabla_{\mathbf{x}_t}R(\mathbf{x}_t). \tag{11}$$

Here, $\eta$ is a guidance scale. Since we lack an explicit, differentiable reward model $R(\mathbf{x}_t)$, we draw from the solution to the KL-regularized RL objective (Equation 8) and from Rafailov et al. (2024); Zhu et al. (2025) to define an implicit, time-dependent reward function $R_t(\mathbf{x}_t)$ that represents the preference learned by $\pi_\theta$ throughout the generative process:

$$R_t(\mathbf{x}_t) \triangleq \beta\log\frac{p_{\theta,t}(\mathbf{x}_t)}{p_{\text{ref},t}(\mathbf{x}_t)}. \tag{12}$$

$p_{\theta,t}$ and $p_{\text{ref},t}$ are the marginal probability distributions of the noisy sample $\mathbf{x}_t$ under the RL-aligned and reference models, respectively, and $\beta$ is the KL-coefficient from the original RL fine-tuning objective.

---

**Algorithm 1** Sampling with Reinforcement Learning Guidance (RLG)

---

1: **Input:** Pre-trained model velocity $\mathbf{v}_{\text{ref}}$, RL-finetuned model velocity $\mathbf{v}_{\text{RL}}$, condition $c$, RLG scale $w$, number of steps $N$.
2: Sample initial noise $\mathbf{x}_1 \sim \mathcal{N}(0, \mathbf{I})$.
3: **for** $t = 1, \ldots, N$ **do**
4:     Compute reference velocity: $\mathbf{v}_{\text{ref},t} = \mathbf{v}_{\text{ref}}(\mathbf{x}_t, t|c)$.
5:     Compute RL-aligned velocity: $\mathbf{v}_{\text{RL},t} = \mathbf{v}_{\text{RL}}(\mathbf{x}_t, t|c)$.
6:     Compute the guided velocity using RLG:
7:     $\hat{\mathbf{v}}_{\text{RLG},t} = (1-w)\mathbf{v}_{\text{ref},t} + w\mathbf{v}_{\text{RL},t}$.
8:     Update the sample using a chosen ODE solver step:
9:     $\mathbf{x}_{t+1} = \text{SolverStep}(\mathbf{x}_t, \hat{\mathbf{v}}_{\text{RLG},t})$.
10: **end for**
11: **Return:** Generated sample $\mathbf{x}_{N+1}$.

---

To use this implicit reward for guidance, we take its gradient with respect to $\mathbf{x}_t$, yielding a simple result:

$$\nabla_{\mathbf{x}_t} R_t(\mathbf{x}_t) = \beta \left[ \nabla_{\mathbf{x}_t} \log p_{\theta,t} - \nabla_{\mathbf{x}_t} \log p_{\text{ref},t} \right]$$
$$= \beta \left[ \mathbf{s}_\theta(\mathbf{x}_t, t) - \mathbf{s}_{\text{ref}}(\mathbf{x}_t, t) \right]. \tag{13}$$

Substituting this into Equation 9 yields the general formula for reward gradient guidance:

$$\hat{\mathbf{s}}_{\text{RLG}}(\mathbf{x}_t, t) = (1-w)\mathbf{s}_{\text{ref}} + w\mathbf{s}_\theta, \tag{14}$$

which is a linear interpolation of score functions, interpretable as implicit reward gradient guidance. Using Eq. 2, the same applies to velocity fields, yielding:

$$\hat{\mathbf{v}}_{\text{RLG}}(\mathbf{x}_t, t) = (1-w)\mathbf{v}_{\text{ref}}(\mathbf{x}_t, t) + w\mathbf{v}_\theta(\mathbf{x}_t, t), \tag{15}$$

where $w$ is the RLG guidance scale. A value of $w = 0$ recovers the original model, $w = 1$ recovers the RL-finetuned model, and $w > 1$ extrapolates the learned alignment. The full sampling procedure is outlined in Algorithm 1.

THEORETICAL JUSTIFICATION: RLG AS KL-COEFFICIENT CONTROL

RLG's mechanisms can be explained by a complementary theoretical justification. Similar to CFG Ho & Salimans (2022), the guided score $\hat{\mathbf{s}}_{\text{RLG}}$ corresponds to sampling from a new time-dependent distribution:

$$\hat{\mathbf{s}}_{\text{RLG}} = \nabla_{\mathbf{x}_t} \log \left( p_{\text{ref},t}(\mathbf{x}_t)^{1-w} p_{\theta,t}(\mathbf{x}_t)^w \right). \tag{16}$$

As $t \to 0$, the noisy sample $\mathbf{x}_t$ approaches the clean data $\mathbf{x}_0$. In this limit, the marginal distributions $p_{\text{ref},t}$ and $p_{\theta,t}$ converge to their corresponding final distributions, $p_{\text{ref}}(\mathbf{x}_0)$ and $p_\theta(\mathbf{x}_0)$. Therefore, the score function guiding the final steps of generation points towards a target distribution $\hat{p}_{\text{RLG}}(\mathbf{x}_0)$ of the form: $p_{\text{ref}}(\mathbf{x}_0)^{1-w} p_\theta(\mathbf{x}_0)^w$.

Assuming the RL-finetuned model $\pi_\theta$ has converged to the optimal distribution from Rafailov et al. (2024; 2023) (i.e., $p_\theta(\mathbf{x}_0) \propto p_{\text{ref}}(\mathbf{x}_0) \exp(\frac{1}{\beta} R(\mathbf{x}_0))$), we can substitute this into the expression for the RLG distribution:

$$\hat{p}_{\text{RLG}}(\mathbf{x}_0) \propto p_{\text{ref}}(\mathbf{x}_0)^{1-w} \left( p_{\text{ref}}(\mathbf{x}_0) \exp\left( \frac{1}{\beta} R(\mathbf{x}_0) \right) \right)^w$$
$$\propto p_{\text{ref}}(\mathbf{x}_0) \exp\left( \frac{1}{\beta/w} R(\mathbf{x}_0) \right). \tag{17}$$

This derivation reveals a crucial insight: RLG with guidance scale $w$ is mathematically equivalent to sampling from the optimal policy of an RL objective with an effective KL-regularization coefficient of $\beta/w$.

We empirically validated this result with a small-scale demonstration. Our experimental setting uses a flow matching model defined on the real line, with a pretrained target (base) Gaussian mixture

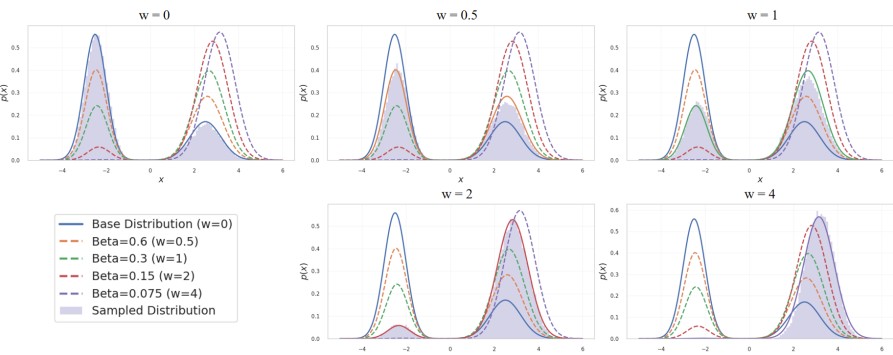

Figure 2: Small-scale demonstration supporting the theoretical justification of RLG. Each subplot shows the sampled distribution under a different RLG weight $w$, while the curves represent the corresponding theoretically predicted RL-fine-tuned distributions. Here, $\beta$ denotes the KL regularization coefficient.

distribution: $p_{\text{base}}(x) \triangleq 0.7\mathcal{N}(-2.5, 0.25) + 0.3\mathcal{N}(2.5, 0.49)$. The reward function is $r(x) = 0.1x$. We fine-tuned the pretrained model using a policy gradient algorithm with a KL coefficient $\beta = 0.3$, a batch size of 64, and a learning rate of $1 \times 10^{-5}$. Figure 2 presents sampled distributions under various RLG weights $w$, alongside the corresponding theoretical distribution curves $p_{\text{rl}}(x) \propto p_{\text{base}}(x) \exp\left(\frac{1}{\beta} r(x)\right)$. Results show that RLG-sampled distributions closely match theoretically predicted RL targets, corroborating our analysis.

Selecting $w > 1$ dynamically reduces the regularization penalty at inference time, allowing the model to pursue higher rewards more aggressively than the original RL-fine-tuned model. Conversely, $w < 1$ increases regularization. This provides principled justification for RLG's capacity to extrapolate or interpolate beyond the learned policy, offering a powerful and theoretically grounded mechanism to control the trade-off between alignment and fidelity.

## 4 EXPERIMENTS

This section empirically validates RLG's effectiveness. Experiments are conducted on various text-to-image (T2I) alignment tasks. In each case, the original pre-trained model serves as $\mathbf{v}_{\text{ref}}$, and the RL-aligned model as $\mathbf{v}_\theta$. RLG is applied as described in Equation 15. For all experiments, sampling steps were set to 20. A comprehensive list of additional hyperparameters can be found in the appendix F.1.

### 4.1 RLG UNIVERSALLY ENHANCES ALIGNMENT ACROSS DIVERSE TASKS

A key strength of RLG is its broad applicability. This training-free method consistently enhances model capabilities across diverse alignment tasks, from high-level compositional understanding to fine-grained subject fidelity.

**Structured Generation: Compositionality and Text Rendering.** RLG is first evaluated on tasks requiring precise adherence to structured prompts. For compositional generation, the GenEval benchmark Ghosh et al. (2023) is used with a GRPO-finetuned SD3.5-M model Liu et al. (2025b). This task tests the model's ability to correctly render object relationships, counts, and attributes. Following the benchmark's protocol, a Mask2Former model Cheng et al. (2022) verifies the presence and properties of objects specified in prompts within the official GenEval test set.

For visual text rendering, an GRPO-finetuned SD3.5-M model on the Optical Character Recognition (OCR) task Mori et al. (1999) (also from Liu et al. (2025b)) measures its ability to accurately render text from prompts that contain the exact string to appear in the image. OCR accuracy is calculated based on the normalized edit distance $(1 - d_{\text{norm}})$, where $d_{\text{norm}}$ is the Levenshtein distance Yujian &

Table 1: Quantitative results for Human Preference Alignment. Mean scores for Aesthetic Score, ImageReward, and PickScore are reported. For each metric, values after the slash (/) indicate win rates (%) against the standard RL-finetuned model ($w_{\mathrm{RL}} = 1.0$). SD denotes stable-diffusion models Rombach et al. (2022).

| Model | RL | $w_{\mathrm{RL}}$ | Aesthetic Score (↑) | ImageReward (↑) | PickScore (↑) |
|---|---|---|---|---|---|
| **SD1.5** | **DPO** Lee et al. (2023) | 0.0 | 5.51 / 38.48% | -0.02 / 36.67% | 20.03 / 27.34% |
| | | 1.0 | 5.61 | 0.20 | 20.39 |
| | | 1.4 | 5.63 / 53.56% | 0.25 / 53.47% | 20.46 / 57.86% |
| | | 2.4 | **5.64 / 56.25%** | **0.32 / 57.08%** | **20.56 / 61.23%** |
| **SDXL-base** | **SPO** Liang et al. (2024) | 0.0 | 6.10 / 17.87% | 0.72 / 18.02% | 21.66 / 7.62% |
| | | 1.0 | 6.42 | 1.12 | 22.69 |
| | | 1.2 | 6.45 / 59.81% | 1.13 / 54.10% | **22.71 / 54.64%** |
| | | 1.4 | **6.48 / 62.99%** | **1.14 / 54.15%** | 22.71 / 54.35% |
| **SD3.5-M** | **GRPO** Liu et al. (2025b) | 0.0 | 5.97 / 11.33% | 0.99 / 17.29% | 21.75 / 2.39% |
| | | 1.0 | 6.45 | 1.40 | 23.29 |
| | | 1.4 | 6.54 / 69.28% | **1.40 / 54.44%** | 23.48 / **74.95%** |
| | | 2.2 | **6.64 / 77.39%** | 1.39 / 53.56% | **23.58** / 73.68% |

Table 2: Pperformance on the GenEval benchmark. We report accuracy (%) for each compositional sub-task and the overall average score across different RLG guidance scales ($w_{\mathrm{RL}}$).

| Model | $w_{\mathrm{RL}}$ | Single Obj. | Two Objs. | Colors | Color Attr. | Counting | Position | Overall Score |
|---|---|---|---|---|---|---|---|---|
| **SD3.5-M** | 0.0 | 97.81 | 80.56 | 80.05 | 51.75 | 53.44 | 23.00 | 64.44 |
| | 1.0 | **100.00** | 98.99 | 89.63 | 84.00 | 92.81 | 93.75 | 93.20 |
| | 1.2 | 99.69 | 98.74 | 90.69 | 86.11 | 92.81 | 94.25 | 93.72 |
| | 1.4 | 99.69 | **99.24** | 91.49 | **86.50** | **94.69** | 94.50 | **94.35** |
| | 1.6 | **100.00** | 98.99 | **91.76** | 86.00 | 93.75 | **95.00** | 94.25 |

Bo (2007) between generated text (extracted using PaddleOCR Authors (2020)) and the ground-truth text, normalized by ground-truth text length.

Tables 2, 3 and Figures 11, 8 show that while RL-finetuned models ($w_{\mathrm{RL}} = 1.0$) already substantially improve over their base counterparts, extrapolating with RLG unlocks further significant gains. On GenEval, RLG pushes the overall score from 93.20% to a peak of **94.35%**, improving almost all key compositional tasks. On the OCR task, RLG boosts accuracy from 88.6% to a new state-of-the-art of **93.0%** with minimal impact on aesthetic score. These results confirm RLG effectively amplifies the model's learned ability to follow complex structural constraints.

Table 3: Quantitative results for the visual text rendering task. This table shows the Optical Character Recognition (OCR) accuracy at different RLG guidance scales.

| Model | $w_{\mathrm{RL}}$ | OCR Acc (↑) | Aesthetic Score (↑) |
|---|---|---|---|
| SD3.5-M | 0.0 | 0.543 | 5.40 |
| | 0.4 | 0.785 | 5.28 |
| | 0.6 | 0.838 | 5.25 |
| | 1.0 | 0.886 | 5.20 |
| | 1.2 | 0.894 | 5.17 |
| | 1.6 | 0.910 | 5.13 |
| | 2.2 | 0.921 | 5.07 |
| | 2.8 | **0.930** | 5.00 |

**Fidelity-Driven Generation: Inpainting and Personalization.** We next test RLG on tasks demanding high fidelity to reference content: image inpainting and personalized generation. For image inpainting, we use the PrefPaint Bui et al. (2025) model, an RL-finetuned model built on stable-diffusion-inpainting Podell et al. (2023) and designed to fill masked regions according to human

preferences. To evaluate quality, we use Preference Reward metrics Bui et al. (2025) on the dataset detailed in appendix I. For personalized generation, we use PatchDPO Huang et al. (2025), an RL-finetuned model optimized to maintain subject identity from reference images. Here, the original pre-trained model (IP-Adapter-Plus Ye et al. (2023)) serves as the base ($\mathbf{v}_{\text{ref}}$), and PatchDPO as the RL-aligned model ($\mathbf{v}_\theta$). Subject fidelity is measured using two standard image-similarity metrics: CLIP-I Ruiz et al. (2023) and DINO Caron et al. (2021), evaluated on the DreamBench Ruiz et al. (2023) benchmark, detailed in appendix J.

The results, summarized in Table 4, again show RLG's effectiveness. For inpainting, RLG pushes the preference score beyond the original PrefPaint model, peaking at **0.368**. For personalized generation, RLG further refines subject fidelity, increasing the DINO score to **0.730** and CLIP-I score to **0.843**. In both cases, RLG provides measurable enhancement over state-of-the-art RL-finetuned models without any additional training.

Table 4: RLG enhances performance on distinct fidelity-driven tasks. The evaluation metrics are presented separately for each task.

| *Task: Image Inpainting* | |
| --- | --- |
| **Method** | **Pref. Reward ($\uparrow$)** |
| Base ($w_{\text{RL}} = 0$) | 0.080 |
| PrefPaint ($w_{\text{RL}} = 1.0$) | 0.358 |
| RLG ($w_{\text{RL}} = 1.2$) | 0.367 |
| RLG ($w_{\text{RL}} = 1.4$) | **0.368** |
| RLG ($w_{\text{RL}} = 1.6$) | 0.366 |

| *Task: Personalized Generation* | | |
| --- | --- | --- |
| **Method** | **DINO ($\uparrow$)** | **CLIP-I ($\uparrow$)** |
| IP-Adapter-Plus ($w_{\text{RL}} = 0$) | 0.692 | 0.826 |
| PatchDPO ($w_{\text{RL}} = 1.0$) | 0.724 | 0.839 |
| RLG ($w = 1.2$) | **0.730** | 0.841 |
| RLG ($w = 1.8$) | 0.730 | **0.843** |

## 4.2 RLG is Effective Across Diverse RL Algorithms and Model Architectures

To demonstrate RLG's broad applicability and model-agnostic nature, we evaluate its consistent enhancement of models differing in generative architecture (i.e., standard diffusion vs. modern flow matching) and the specific reinforcement learning algorithm used for their initial alignment. This also extends to algorithms such as GRPO, whose optimal policy does not necessarily conform to Equation 8.

**Experimental Setup.** We analyze RLG on the human preference alignment task, leveraging three distinct, publicly available RL-finetuned models, each representing a unique combination of architecture and alignment method:

- **SD1.5 + DPO:** A Stable Diffusion v1.5 model Rombach et al. (2022) aligned using Direct Preference Optimization (DPO) Wallace et al. (2024).
- **SDXL + SPO:** A Stable Diffusion XL model Podell et al. (2023) aligned using Step-wise Preference Optimization (SPO) Liang et al. (2024).
- **SD3.5-M + GRPO:** A Stable Diffusion 3.5 Medium flow matching model Esser et al. (2024) aligned using Group Relative Policy Optimization (GRPO) Liu et al. (2025b).

For evaluation, we use three established automated reward models: Aesthetic Score, ImageReward, and PickScore, with details provided in the appendix.

**Results.** Table 1 summarizes quantitative results, unequivocally demonstrating that RLG consistently delivers a significant performance boost across all configurations. The effect is particularly pronounced on the state-of-the-art GRPO-tuned SD3.5-M flow model, where RLG achieves a **74.95%** win rate on PickScore against the original finetuned model ($w_{\text{RL}} = 1.0$). As visually confirmed in Figures 1, 5, 6 and 7, increasing the RLG scale consistently enhances image detail and aesthetic appeal.

## 4.3 RLG Enables Flexible Control Over Alignment Strength

Standard RL fine-tuning fixes alignment strength, offering no inference-time flexibility. In contrast, RLG dynamically controls alignment strength with a powerful, training-free mechanism.

Table 5: RLG provides dynamic control over image compressibility. RLG allows for both interpolation and extrapolation beyond the original RL-tuned model's capability ($w_{\text{RL}} = 1.0$).

| Task | $w_{\text{RL}}$ | Compression Ratio |
|---|---|---|
| Low Compressibility | 0.4 | 1.14 |
| | 0.6 | 1.22 |
| | 1.0 | 1.35 |
| | 1.6 | **1.43** |
| | 3.0 | 1.37 |
| High Compressibility | 0.4 | 0.75 |
| | 0.6 | 0.64 |
| | 1.0 | 0.45 |
| | 1.6 | 0.18 |
| | 2.2 | **0.17** |

**Controlling a Fundamental Property: Image Compressibility.** We first demonstrate RLG's control over image compressibility, a low-level property. We used two DDPO-finetuned SD1.4 models Black et al. (2023) to reward either high or low image compressibility. Standard RL produces a model with fixed alignment; for instance, the low-compressibility model is locked at a 1.35 compression ratio in average (where $w_{\text{RL}} = 1.0$). RLG transforms this static point into a dynamic spectrum. As shown in Table 5, users can weaken alignment by setting $w_{\text{RL}} < 1.0$ (e.g., achieving a 1.14 ratio) or intensify it with $w_{\text{RL}} > 1.0$, pushing the ratio beyond the fine-tuned limit to a peak of **1.43**. RLG thus provides an inference-time 'slider' for alignment strength, a capability static fine-tuning lacks.

**Balancing Competing Objectives: Text Accuracy vs. Aesthetics.** Maximizing one alignment objective often compromises another. Table 3 illustrates this conflict. The standard RL-finetuned model ($w_{\text{RL}} = 1.0$) achieves 88.6% OCR accuracy, but its Aesthetic Score is fixed at 5.20. This trade-off is unalterable. RLG transforms this static outcome into flexible control. For instance, users prioritizing aesthetics over maximum text accuracy can set $w_{\text{RL}} < 1.0$. Conversely, others can push for peak accuracy at the cost of aesthetics by setting $w_{\text{RL}} = 2.8$. This inference-time flexibility for users to choose their preferred sweet spot on the trade-off curve is a key advantage RLG holds over static fine-tuning.

## 5 CONCLUSION AND DISCUSSION

In this paper, we proposed Reinforcement Learning Guidance (RLG), a training-free method for dynamically controlling generative model alignment at inference. By interpolating or extrapolating beyond trained preferences, RLG effectively modulates the KL-regularization penalty to pursue higher rewards. Extensive experiments show consistent gains across diverse tasks, making RLG a simple yet powerful control layer over learned preferences.

Despite its broad empirical success, RLG has limitations that motivate future work. First, inheriting from CFG, RLG shares its fundamental drawback: CFG-based sampling does not guarantee approximation to the target marginal distribution (Bradley & Nakkiran, 2024; Skreta et al., 2025). Thus it exhibits inherent flaws in the subsequent analytical derivations. Second, our theory linking the RLG scale $w$ to the KL coefficient $\beta$ assumes convergence to the optimal policy under a standard reward–KL objective. This is an idealized assumption and only holds when the optimization objective is a standard mixture of expected return and KL divergence. For methods such as GRPO, the optimal policy does not adhere to this theoretical form (Vojnovic & Yun, 2025). Finally, future work could explore adaptive RLG scales that vary across timesteps or combine RLG with other orthogonal control methods to achieve even more nuanced generation.

## 6 REPRODUCIBILITY STATEMENT

To ensure reproducibility, we provide the complete implementation of RLG in an anonymous repository at https://anonymous.4open.science/r/

Reinforcement-learning-guidance-7B5A/. The sources of all baseline models and benchmark datasets are detailed in their respective subsections within Section 4. For reference, Table 6 consolidates the information and sources of all models utilized in this study.

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

## A THE USE OF LARGE LANGUAGE MODELS (LLMS)

Large Language Models were used solely for language polishing and writing refinement of this manuscript, including grammar correction and clarity improvement. All research content, methodology, analysis, and conclusions are entirely the original work of the authors.

## B MODEL SPECIFICATION

The following table 6 lists the base model, the RL-finetuned model, the reward models and their corresponding links.

## C THEORETICAL DERIVATIONS

This appendix provides the formal derivations discussed in the main paper.

### C.1 PROOF OF THE OPTIMAL POLICY FOR KL-REGULARIZED RL

We aim to find the policy $\pi^*$ that solves the optimization problem defined in Equation 7:

$$\pi^* = \arg\max_{\pi} \left( \mathbb{E}_{\mathbf{x} \sim \pi(\mathbf{x})}[R(\mathbf{x})] - \beta D_{\mathrm{KL}}(\pi(\mathbf{x}) \| \pi_{\mathrm{ref}}(\mathbf{x})) \right) \tag{18}$$

subject to the constraint that $\pi(\mathbf{x})$ is a valid probability distribution, i.e., $\int \pi(\mathbf{x})d\mathbf{x} = 1$.

First, we expand the objective functional $J(\pi)$:

$$J(\pi) = \int \pi(\mathbf{x})R(\mathbf{x})d\mathbf{x} - \beta \int \pi(\mathbf{x}) \log \frac{\pi(\mathbf{x})}{\pi_{\mathrm{ref}}(\mathbf{x})} d\mathbf{x}$$

$$= \int \left( \pi(\mathbf{x})R(\mathbf{x}) - \beta\pi(\mathbf{x}) \log \pi(\mathbf{x}) + \beta\pi(\mathbf{x}) \log \pi_{\mathrm{ref}}(\mathbf{x}) \right) d\mathbf{x} \tag{19}$$

This is a constrained optimization problem that can be solved using the calculus of variations with a Lagrange multiplier, $\lambda$, for the probability distribution constraint. The Lagrangian is:

$$\mathcal{L}(\pi, \lambda) = J(\pi) + \lambda \left( \int \pi(\mathbf{x})d\mathbf{x} - 1 \right) \tag{20}$$

To find the optimal policy $\pi^*$, we take the functional derivative of $\mathcal{L}$ with respect to $\pi(\mathbf{x})$ and set it to zero.

$$\frac{\delta\mathcal{L}}{\delta\pi(\mathbf{x})} = \frac{\partial}{\partial\pi(\mathbf{x})} \left[ \pi R - \beta\pi \log \pi + \beta\pi \log \pi_{\mathrm{ref}} + \lambda\pi \right] = 0$$

$$= R(\mathbf{x}) - \beta(\log \pi(\mathbf{x}) + 1) + \beta \log \pi_{\mathrm{ref}}(\mathbf{x}) + \lambda = 0 \tag{21}$$

Table 6: Models and Their Corresponding Links.

| Model/Reward Function | Link |
| --- | --- |
| SD3.5-M Esser et al. (2024) | `https://huggingface.co/stabilityai/stable-diffusion-3.5-medium` |
| SD1.5 Rombach et al. (2022) | `https://huggingface.co/stable-diffusion-v1-5/stable-diffusion-v1-5` |
| SD1.4 Rombach et al. (2022) | `https://huggingface.co/CompVis/stable-diffusion-v1-4` |
| SDXL-base Podell et al. (2023) | `https://huggingface.co/stabilityai/stable-diffusion-xl-base-1.0` |
| SD3.5M-FlowGRPO-PickScore Liu et al. (2025b) | `https://huggingface.co/jieliu/SD3.5M-FlowGRPO-PickScore` |
| SD3.5M-FlowGRPO-Text Liu et al. (2025b) | `https://huggingface.co/jieliu/SD3.5M-FlowGRPO-Text` |
| SD3.5M-FlowGRPO-GenEval Liu et al. (2025b) | `https://huggingface.co/jieliu/SD3.5M-FlowGRPO-GenEval` |
| dpo-sd1.5-text2image-v1 Lee et al. (2023) | `https://huggingface.co/mhdang/dpo-sd1.5-text2image-v1` |
| SPO-SDXL-4k-p-10ep Liang et al. (2024) | `https://huggingface.co/SPO-Diffusion-Models/SPO-SDXL_4k-p_10ep` |
| dpo-sdxl-text2image-v1 Lee et al. (2023) | `https://huggingface.co/mhdang/dpo-sdxl-text2image-v1` |
| ddpo-compressibility Black et al. (2023) | `https://huggingface.co/kvablack/ddpo-compressibility` |
| ddpo-incompressibility Black et al. (2023) | `https://huggingface.co/kvablack/ddpo-incompressibility` |
| Aesthetic Score Schuhmann & Beaumont (2021) | `https://github.com/LAION-AI/aesthetic-predictor` |
| ImageReward Xu et al. (2023) | `https://huggingface.co/THUDM/ImageReward` |
| PickScore Kirstain et al. (2023) | `https://huggingface.co/yuvalkirstain/PickScore_v1` |
| clip-vit-large-patch14 Radford et al. (2021) | `https://huggingface.co/openai/clip-vit-large-patch14` |
| stable-diffusion-inpainting Podell et al. (2023) | `https://aihub.caict.ac.cn/models/runwayml/stable-diffusion-inpainting` |
| prefpaint Bui et al. (2025) | `https://huggingface.co/kd5678/prefpaint-v1.0` |
| prefpaintreward Bui et al. (2025) | `https://huggingface.co/kd5678/prefpaintReward` |
| IP-Adapter-Plus Ye et al. (2023) | `https://huggingface.co/h94/IP-Adapter` |
| PatchDPO Huang et al. (2025) | `https://huggingface.co/hqhQAQ/PatchDPO` |

Now, we solve for $\log \pi(\mathbf{x})$:

$$\beta \log \pi(\mathbf{x}) = R(\mathbf{x}) + \beta \log \pi_{\text{ref}}(\mathbf{x}) + \lambda - \beta$$

$$\log \pi(\mathbf{x}) = \frac{1}{\beta} R(\mathbf{x}) + \log \pi_{\text{ref}}(\mathbf{x}) + \frac{\lambda - \beta}{\beta} \tag{22}$$

Exponentiating both sides gives the form of the optimal policy $\pi^*(\mathbf{x})$:

$$\pi^*(\mathbf{x}) = \exp\left(\frac{1}{\beta} R(\mathbf{x}) + \log \pi_{\text{ref}}(\mathbf{x}) + \frac{\lambda - \beta}{\beta}\right)$$

$$= \pi_{\text{ref}}(\mathbf{x}) \exp\left(\frac{1}{\beta} R(\mathbf{x})\right) \exp\left(\frac{\lambda - \beta}{\beta}\right) \tag{23}$$

The term $\exp\left(\frac{\lambda - \beta}{\beta}\right)$ is a constant that does not depend on $\mathbf{x}$. This constant serves as the normalization factor to ensure that $\int \pi^*(\mathbf{x}) d\mathbf{x} = 1$. Let us denote this normalization constant as $1/Z(\beta)$. Therefore, the optimal distribution is:

$$\pi^*(\mathbf{x}) = \frac{1}{Z(\beta)} \pi_{\text{ref}}(\mathbf{x}) \exp\left(\frac{1}{\beta} R(\mathbf{x})\right) \tag{24}$$

This is equivalent to the proportional relationship given in Equation 8:

$$\pi^*(\mathbf{x}) \propto \pi_{\text{ref}}(\mathbf{x}) \exp\left(\frac{1}{\beta} R(\mathbf{x})\right) \tag{25}$$

This completes the proof.

### C.2 EQUIVALENCE OF THE DPO OBJECTIVE

The Direct Preference Optimization (DPO) framework is derived by re-parameterizing the KL-regularized RL objective in terms of preferences, thereby avoiding the need to explicitly train a reward model. The derivation shows that optimizing the DPO loss is equivalent to optimizing the policy towards the same theoretical distribution $\pi^*$ derived above.

The derivation proceeds as follows:

1. **Express Reward in terms of Policies:** We start with the optimal policy solution from the previous section and rearrange it to solve for the reward function $R(\mathbf{x})$:

$$\pi^*(\mathbf{x}) = \frac{1}{Z(\beta)} \pi_{\text{ref}}(\mathbf{x}) \exp\left(\frac{1}{\beta} R(\mathbf{x})\right)$$

$$\implies R(\mathbf{x}) = \beta \log\left(\frac{\pi^*(\mathbf{x})}{\pi_{\text{ref}}(\mathbf{x})}\right) + \beta \log Z(\beta) \tag{26}$$

The term $\beta \log Z(\beta)$ is a constant with respect to $\mathbf{x}$.

2. **Model Human Preferences:** Human preferences are typically collected as pairs $(\mathbf{x}_w, \mathbf{x}_l)$, where $\mathbf{x}_w$ is preferred over $\mathbf{x}_l$. The Bradley-Terry model maps reward scores to preference probabilities:

$$p(\mathbf{x}_w \succ \mathbf{x}_l) = \sigma(R(\mathbf{x}_w) - R(\mathbf{x}_l)) \tag{27}$$

where $\sigma(\cdot)$ is the sigmoid function.

3. **Combine Reward and Preference Models:** We substitute the policy-based expression for the reward into the Bradley-Terry model. The constant term $\beta \log Z(\beta)$ cancels out perfectly:

$$R(\mathbf{x}_w) - R(\mathbf{x}_l)$$

$$= \left(\beta \log \frac{\pi^*(\mathbf{x}_w)}{\pi_{\text{ref}}(\mathbf{x}_w)} + C\right) - \left(\beta \log \frac{\pi^*(\mathbf{x}_l)}{\pi_{\text{ref}}(\mathbf{x}_l)} + C\right)$$

$$= \beta \left(\log \frac{\pi^*(\mathbf{x}_w)}{\pi_{\text{ref}}(\mathbf{x}_w)} - \log \frac{\pi^*(\mathbf{x}_l)}{\pi_{\text{ref}}(\mathbf{x}_l)}\right) \tag{28}$$

Thus, the ground-truth preference probability can be expressed entirely in terms of the optimal policy $\pi^*$ and the reference policy $\pi_{\text{ref}}$:

$$p(\mathbf{x}_w \succ \mathbf{x}_l) = \sigma\left(\beta\left(\log\frac{\pi^*(\mathbf{x}_w)}{\pi_{\text{ref}}(\mathbf{x}_w)} - \log\frac{\pi^*(\mathbf{x}_l)}{\pi_{\text{ref}}(\mathbf{x}_l)}\right)\right) \tag{29}$$

4. **Construct the DPO Loss:** DPO seeks to find a policy $\pi_\theta$ that maximizes the log-likelihood of the observed human preferences. This is equivalent to minimizing the negative log-likelihood loss. By replacing the theoretical optimal policy $\pi^*$ with our parameterized model policy $\pi_\theta$, we arrive at the DPO loss function:

$$\mathcal{L}_{\text{DPO}} = -\mathbb{E}_{(\mathbf{x}_w, \mathbf{x}_l) \sim \mathcal{D}}\left(\log p(\mathbf{x}_w \succ \mathbf{x}_l)\right) \tag{30}$$

By minimizing this loss, we are directly training the policy $\pi_\theta$ to satisfy the same mathematical relationship that defines the optimal RL policy $\pi^*$. Therefore, the policy obtained by successfully optimizing the DPO objective, $\pi^*_{DPO}$, converges to the same theoretical distribution as the one found by KL-regularized RL, where the reward function $R(\mathbf{x})$ is implicitly defined by the human preference dataset.

## C.3   DERIVATION OF VELOCITY-SCORE RELATIONSHIP

This section provides a detailed derivation of the relationship between the velocity field $\mathbf{v}(\mathbf{x}, t)$ used in Flow Matching models and the score function $\mathbf{s}(\mathbf{x}, t)$ used in Denoising Diffusion Models, as stated in Equation 2.

The unifying perspective relies on a common reference path $(X_t)_{t \in [0,1]}$ that interpolates between an initial noise variable $X_1 \sim p_1 = \mathcal{N}(0, \mathbf{I})$ and a data sample $X_0 \sim p_{\text{data}}$. This path is defined by linear interpolation:

$$X_t = \beta_t X_1 + \alpha_t X_0 \tag{31}$$

where $\alpha_t$ and $\beta_t$ are scalar functions of time, with $\alpha_0 = \beta_1 = 0$ and $\alpha_1 = \beta_0 = 1$.

In Denoising Diffusion Models, the model learns to predict the score function $\mathbf{s}(\mathbf{X}_t, t) = \nabla_{\mathbf{X}_t} \log p_t(\mathbf{X}_t)$. For the chosen linear interpolation path where $X_1 \sim \mathcal{N}(0, \mathbf{I})$, it's a known property that the score function is related to the conditional expectation of $X_0$ and $X_1$ given $X_t$. Specifically, the optimal denoised estimate of $X_0$, denoted $\hat{X}_0(\mathbf{X}_t, t)$, and the optimal estimate of the noise $X_1$, denoted $\hat{X}_1(\mathbf{X}_t, t)$, can be expressed in terms of $X_t$ and its score:

$$\hat{X}_0(\mathbf{X}_t, t) = \mathbb{E}[X_0|X_t] = \frac{1}{\alpha_t}(\mathbf{X}_t + \beta_t^2 \mathbf{s}(\mathbf{X}_t, t)) \tag{32}$$

$$\hat{X}_1(\mathbf{X}_t, t) = \mathbb{E}[X_1|X_t] = -\beta_t \mathbf{s}(\mathbf{X}_t, t) \tag{33}$$

These relationships hold under the assumption that the conditional distribution $p(X_t|X_0)$ is a Gaussian $X_t \sim \mathcal{N}(\alpha_t X_0, \beta_t^2 \mathbf{I})$, which is implied by the path definition with $X_1 \sim \mathcal{N}(0, \mathbf{I})$.

For Flow Matching models, the objective is to learn a velocity field $\mathbf{v}(\mathbf{X}_t, t)$ that describes the deterministic trajectory of samples via an ordinary differential equation $d\mathbf{X}_t = \mathbf{v}(\mathbf{X}_t, t)dt$. This velocity field matches the time derivative of the reference flow, $\frac{d}{dt}X_t$. Differentiating Equation 31 with respect to time $t$:

$$\mathbf{v}(\mathbf{X}_t, t) = \frac{d}{dt}X_t = \dot{\beta}_t X_1 + \dot{\alpha}_t X_0 \tag{34}$$

To express the velocity field in terms of the current state $\mathbf{X}_t$ and the score function $\mathbf{s}(\mathbf{X}_t, t)$, we substitute the expressions for $\hat{X}_0(\mathbf{X}_t, t)$ (Equation 32) and $\hat{X}_1(\mathbf{X}_t, t)$ (Equation 33) into Equation 34:

$$\mathbf{v}(\mathbf{X}_t, t) = \dot{\beta}_t(-\beta_t \mathbf{s}(\mathbf{X}_t, t)) + \dot{\alpha}_t\left(\frac{1}{\alpha_t}(\mathbf{X}_t + \beta_t^2 \mathbf{s}(\mathbf{X}_t, t))\right)$$

$$= -\dot{\beta}_t \beta_t \mathbf{s}(\mathbf{X}_t, t) + \frac{\dot{\alpha}_t}{\alpha_t}\mathbf{X}_t + \frac{\dot{\alpha}_t}{\alpha_t}\beta_t^2 \mathbf{s}(\mathbf{X}_t, t)$$

Rearranging the terms by grouping $\mathbf{X}_t$ and $\mathbf{s}(\mathbf{X}_t, t)$:

$$\mathbf{v}(\mathbf{X}_t, t) = \left(\frac{\dot{\alpha}_t}{\alpha_t}\right)\mathbf{X}_t + \left(\frac{\dot{\alpha}_t}{\alpha_t}\beta_t^2 - \dot{\beta}_t\beta_t\right)\mathbf{s}(\mathbf{X}_t, t)$$

$$= \left(\frac{\dot{\alpha}_t}{\alpha_t}\right)\mathbf{X}_t + \beta_t\left(\frac{\dot{\alpha}_t}{\alpha_t}\beta_t - \dot{\beta}_t\right)\mathbf{s}(\mathbf{X}_t, t) \tag{35}$$

This derivation confirms Equation 2 from the main paper, establishing the precise mathematical connection between the velocity field learned by Flow Matching and the score function predicted by Denoising Diffusion Models under the common linear interpolation path.

## D    DERIVATION OF THE IMPLICIT TIME-DEPENDENT REWARD

To formalize this, we first need to establish that for any given generative model policy $\pi_\theta$ (represented by its distribution $p_{\theta,t}$), we can define a corresponding reward function for which $\pi_\theta$ is the optimal policy. This concept is well-established in inverse reinforcement learning for discrete MDPs, such as those used for aligning LLMs Sun et al. (2025); Rafailov et al. (2023). We can extend this framework to diffusion models by considering the generation process as a continuous-time MDP Black et al. (2023); Rafailov et al. (2024).

In this diffusion MDP, the state at time $t$ is the noisy sample $\mathbf{x}_t$, and the policy $\pi(\cdot|\mathbf{x}_t)$ determines the transition to the next state $\mathbf{x}_{t-dt}$. Recent theoretical work has shown a bijection between reward functions and optimal Q-functions (and thus optimal policies) in such MDPs. Specifically, following Rafailov et al. (2024), for a given reference policy $\pi_{\text{ref}}$ and a temperature parameter $\beta$, the optimal policy $\pi^*$ for a reward function $r(s_t, a_t)$ satisfies the relationship:

$$\beta \log \frac{\pi^*(a_t|s_t)}{\pi_{\text{ref}}(a_t|s_t)} = r(s_t, a_t) + \Phi(s_{t-dt}) - \Phi(s_t) \tag{36}$$

where $\Phi$ is a potential function, corresponding to the optimal value function $V^*$. This means that any policy $\pi_\theta$ can be viewed as the optimal policy for an implicitly defined reward function, equivalent to the log-policy ratio up to a potential-based shaping term.

By adapting this principle to the continuous state-space of diffusion models, we can define an instantaneous, time-dependent reward function $R_t(\mathbf{x}_t)$ directly in terms of the model's probability density. The policy $\pi_\theta(\cdot|\mathbf{x}_t)$ is governed by the underlying score function $\mathbf{s}_\theta(\mathbf{x}_t, t)$, which itself is the gradient of the log-density $\log p_{\theta,t}(\mathbf{x}_t)$. We can therefore define the implicit reward by relating the marginal densities of the RL-aligned model ($p_{\theta,t}$) and the reference model ($p_{\text{ref},t}$):

$$R_t(\mathbf{x}_t) \triangleq \beta \log \frac{p_{\theta,t}(\mathbf{x}_t)}{p_{\text{ref},t}(\mathbf{x}_t)} \tag{37}$$

Here, $p_{\theta,t}$ is the marginal probability distribution of the noisy image $\mathbf{x}_t$ under the RL-aligned model, and $p_{\text{ref},t}$ is the corresponding distribution for the pre-trained reference model. The parameter $\beta$ is the same KL-regularization coefficient from the original RL objective (Equation 7). This equation defines the reward that the RL-aligned model $\pi_\theta$ is implicitly optimizing for at every point $(\mathbf{x}_t, t)$ in the generation process, relative to the reference model.

## E    ONE-DIMENSIONAL CASE STUDY OF MODEL CONVERGENCE

We further investigate how RLG behaves under different RL algorithms when the RL policy does *not* converge to the ideal reward–reweighted distribution, and why RLG can still improve performance in this realistic regime. To this end, we construct a small-scale one-dimensional case study. Our setting uses a flow-matching model defined on the real line with a pretrained base Gaussian mixture distribution

$$p_{\text{base}}(x) \triangleq 0.7\,\mathcal{N}(-2.5, 0.25) + 0.3\,\mathcal{N}(2.5, 0.49).$$

To bias the model toward a specific region, we introduce a reward function that assigns higher reward to samples near $x = 4$; concretely, we use $r(x) = -0.1|x - 4|$ (up to an additive constant), so that points closer to $x = 4$ receive larger reward. We then fine-tune the pretrained flow with either a

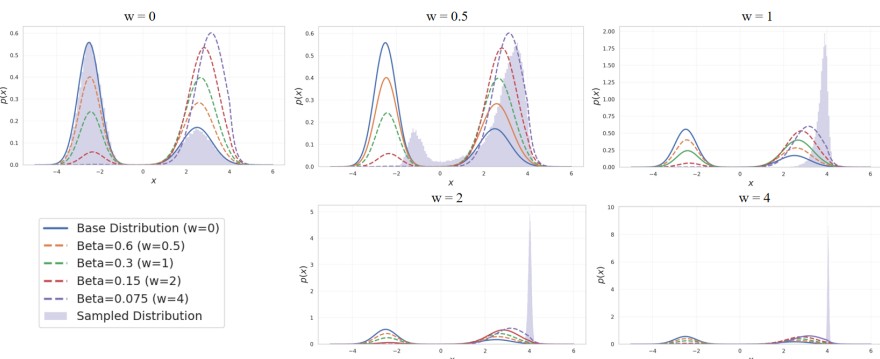

Figure 3: Small-scale demonstration supporting the theoretical justification of RLG. Each subplot shows the sampled distribution under a different RLG weight $w$, while the curves represent the corresponding theoretically predicted RL-fine-tuned distributions. Here, $\beta$ denotes the KL regularization coefficient.

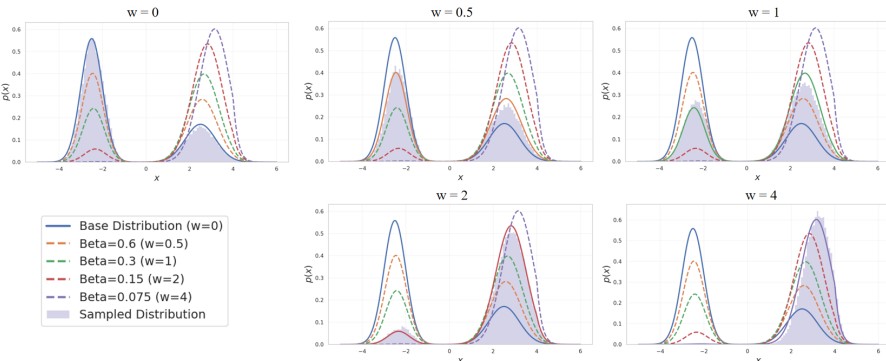

Figure 4: Small-scale demonstration supporting the theoretical justification of RLG. Each subplot shows the sampled distribution under a different RLG weight $w$, while the curves represent the corresponding theoretically predicted RL-fine-tuned distributions. Here, $\beta$ denotes the KL regularization coefficient.

vanilla policy-gradient method or the GRPO algorithm, using a KL coefficient $\beta = 0.3$, batch size 64, and learning rate $1 \times 10^{-5}$, with the pretrained flow as the reference policy.

Figures 3 and 4 show the sampled distributions under different RLG weights $w$, together with the corresponding theoretical reward-reweighted target $p_{\text{rl}}(x) \propto p_{\text{base}}(x) \exp(r(x)/\beta)$. For both the policy-gradient and GRPO experts, increasing $w$ from 0.5 to 1, 2, and 4 consistently shifts probability mass toward the high-reward region around $x = 4$: the left-hand mode is gradually suppressed, and the right-hand mode becomes sharper near the reward peak. In the GRPO case, the standalone RL policy already produces a very sharp and imperfect approximation of the target distribution, yet applying RLG on top of this expert still moves the overall sampler closer to the high-reward region. This toy example illustrates that even when the RL policy does not exactly realize the ideal distribution (as is typical for practical algorithms such as GRPO), RLG with a moderate weight $w$ can reliably steer samples toward higher-reward regions and improve alignment over both the base model and the raw RL-fine-tuned model.

## F    EXPERIMENTAL DETAILS

### F.1    GENERATION HYPER-PARAMETERS

This section provides a detailed overview of the parameters and settings used for the text-to-image generation experiments discussed in the main paper.

To ensure consistency across our evaluations, several parameters were standardized for all models. All images were generated at a resolution of $512 \times 512$ pixels. Text prompts were processed using the models' respective text encoders; For the generative process, we uniformly set the number of sampling steps to 20 for all experiments to maintain a balance between computational cost and output quality.

The Classifier-Free Guidance (CFG) scale, which controls the adherence to the text prompt, was set to the generally recommended value for each model to ensure optimal baseline performance. The specific CFG scales used were:

- **Stable Diffusion 1.4:** A CFG scale of 7.5 was used.
- **Stable Diffusion 1.5:** A CFG scale of 7.5 was used.
- **Stable Diffusion inpainting:** A CFG scale of 7.5 was used.
- **Stable Diffusion XL:** A CFG scale of 5.0 was used.
- **Stable Diffusion 3.5:** A CFG scale of 4.5 was used.

These CFG scales were held constant across all experiments for a given model, allowing for a direct assessment of the impact of our RLG guidance scale ($w$). Other parameters are kept default as diffusers pipeline.

### F.2 Human preference Evaluation Metrics Details

To quantitatively evaluate the performance of our method, we established automated reward models. These models are designed to assess different aspects of image quality and text-to-image alignment, providing a comprehensive evaluation framework.

- **Aesthetic Score**: This metric provides a general assessment of an image's visual appeal. It utilizes a pre-trained CLIP model ('clip-vit-large-patch14') Radford et al. (2021) to extract a feature embedding from the input image. This embedding is then processed by a Multi-Layer Perceptron (MLP) head, loaded with weights provided in Liu et al. (2025b), which regresses the features into a single scalar score, typically on a 1-to-10 scale. The corresponding links are listed in table 6. A higher score indicates a higher predicted aesthetic quality, as judged by human raters in the dataset the MLP was trained on.
- **ImageReward**: Developed by Xu et al. (2023), this is a sophisticated reward model designed to evaluate both the semantic alignment of an image to its text prompt and its overall visual fidelity. Built upon the BLIP-2 architecture Li et al. (2023), it was fine-tuned on a large-scale dataset of human preference feedback, enabling it to serve as a robust, general-purpose proxy for human judgment in text-to-image generation tasks.
- **PickScore**: Introduced by Kirstain et al. (2023), this reward model is specifically trained to predict human preferences based on direct pairwise comparisons. It is derived from the extensive Pick-a-Pic dataset, which contains a vast number of human choices between two images generated from the same prompt. We use the v1 version, which leverages a powerful 'CLIP-ViT-H-14' model Radford et al. (2021). Its strong correlation with human preference makes it particularly relevant for our work.

## G GenEval Benchmark Details

The **GenEval** benchmark provides an automated, object-focused framework for evaluating the compositional capabilities of text-to-image models Ghosh et al. (2023). Unlike holistic metrics that measure overall image quality or text alignment, GenEval offers a fine-grained analysis of a model's ability to adhere to specific compositional instructions within a prompt.

The official test set is comprised of 553 prompts. the prompts are designed to probe several distinct aspects of compositional generation:

- **Single Object:** Tests the model's fundamental ability to render a single specified object.

- **Two Objects:** Assesses the capacity to generate two distinct objects in the same image, testing for co-occurrence.

- **Counting:** Evaluates whether the model can generate a precise number of a given object.

- **Colors:** Measures if an object can be rendered with a specific, designated color.

- **Position:** Tests spatial reasoning by requiring two objects to be placed in a specified relative position (e.g., "a cat to the left of a dog").

- **Attribute Binding:** The most complex task, which requires binding specific attributes (like color) to specific objects (e.g., generating "a red cube and a blue sphere" without swapping attributes).

The evaluation protocol is fully automated, using a sophisticated object detection model to parse the generated images. For our experiments, we adhered to the official methodology, which employs a **Mask2Former** Cheng et al. (2022) model with a Swin-S transformer backbone Liu et al. (2021). This detector identifies objects and verifies their properties and spatial arrangements as dictated by the input prompt.

## H    Image Compressibility Experimental Details

This section provides a detailed description of the experimental setup for the image compressibility task.

**Models and Task Definition.**    The goal of this experiment was to verify that Reinforcement Learning Guidance (RLG) can effectively control a low-level, non-semantic property of generated images: their compressibility. We used the standard Stable Diffusion v1.4 model as our base reference ($\mathbf{v}_{\text{ref}}$). For the aligned models ($\mathbf{v}_\theta$), we utilized two sets of weights from the official DDPO implementation Black et al. (2023):

- **Low Compressibility Model:** We use ddpo-compressibility. This model was fine-tuned to generate images that are less compressible, resulting in larger file sizes when saved in JPEG format. This typically corresponds to images with higher texture detail and complexity.

- **High Compressibility Model:** We use ddpo-incompressibility. This model was fine-tuned to prefer images that are more compressible, resulting in smaller JPEG file sizes. This often corresponds to images with smoother regions and less high-frequency detail.

**Dataset and Prompts.**    Following the DDPO study Black et al. (2023), our evaluation prompts were based on animal classes from the ImageNet dataset. We used 45 distinct animal classes. For each class, we generated 4 images, resulting in a total of 180 images per RLG scale setting. The prompt template used was: "{class_name}".

The 45 animal classes used are: ant, bat, bear, bee, beetle, bird, butterfly, camel, cat, chicken, cow, deer, dog, dolphin, duck, fish, fly, fox, frog, goat, goose, gorilla, hedgehog, horse, kangaroo, lion, lizard, llama, monkey, mouse, pig, rabbit, raccoon, rat, shark, sheep, snake, spider, squirrel, tiger, turkey, turtle, whale, wolf, and zebra.

**Evaluation Metric.**    We evaluated performance using the **Compression Ratio**. For a given prompt, let $\mathbf{x}_{\text{base}}$ be the image generated by the base SD1.4 model, and let $\mathbf{x}_{\text{RLG}}(w)$ be the image generated using RLG with a guidance scale of $w$. Let $S_{jpeg}(\cdot)$ denote the file size of an image after being compressed and saved in JPEG format. The Compression Ratio is defined as:

$$\text{Compression Ratio}(w) = \frac{S_{jpeg}(\mathbf{x}_{\text{RLG}}(w))}{S_{jpeg}(\mathbf{x}_{\text{base}})}$$

The final score reported in Table 5 is the mean of this ratio, averaged across all 180 generated images for each guidance scale. A ratio of 1.0 indicates no change in compressibility compared to the base model.

# I  IMAGE INPAINTING EXPERIMENTAL DETAILS

This section provides a comprehensive overview of the experimental setup, models, and evaluation protocol used for the image inpainting task discussed in the main paper. Our methodology closely follows that of the PrefPaint Bui et al. (2025) study to ensure a fair and direct comparison.

## I.1  MODELS AND TASK DEFINITION

The experiment focuses on conditional image generation for image inpainting, the task of filling in masked (missing) regions of an image in a semantically and visually plausible manner.

- **Base Model ($v_{ref}$):** We use the standard Stable Diffusion inpainting model as our baseline. This model, accessible on diffusers as `runwayml/stable-diffusion-inpainting`, is widely used and serves as the un-aligned reference point for our experiments.

- **RL-Finetuned Model ($v_\theta$):** For the human-preference-aligned expert model, we employ **PrefPaint** Bui et al. (2025). This model is a direct descendant of the base model, which has been further fine-tuned using reinforcement learning. The training process for PrefPaint leveraged a large-scale dataset of over 51,000 human preference judgments on inpainted images, making it an expert policy specialized in generating completions that align with human aesthetic and contextual expectations.

## I.2  EVALUATION DATASET AND PROTOCOL

All quantitative results were generated using the dataset provided by the authors of PrefPaint Bui et al. (2025). The test set was constructed using the following procedure:

1. **Image Sourcing:** A diverse set of high-resolution images was initially sourced from established datasets such as ADE20K and ImageNet. All images were resized to a standard $512 \times 512$ pixel resolution.

2. **Mask Generation:** To simulate realistic inpainting and outpainting scenarios, two distinct masking strategies were employed to create the incomplete images:

   - **Warping Holes (for Inpainting):** This method creates complex, non-rectangular masks inside the image. It simulates the disocclusion that occurs from a slight change in camera viewpoint. A depth map is first estimated for the source image, and then a new virtual camera view is generated with small shifts. The newly visible (disoccluded) regions form the mask that the model must fill. This tests the model's ability to reason about 3D geometry and handle irregular shapes.

   - **Boundary Masks (for Outpainting):** This strategy tests the model's ability to extend a scene beyond its original borders. Masks are created at the edges of the image using two different cropping techniques:
     - *Square Cropping:* A central square region, covering 75% to 85% of the image area, is preserved, masking the outer frame.
     - *Rectangular Cropping:* The full height of the image is preserved, while a central vertical slice, comprising 60% to 65% of the original width, is kept, masking the left and right sides.

## I.3  EVALUATION METRICS

The quality of the generated inpainted images was assessed using the following two automated metrics:

- **Preference Reward:** We use the specialized reward model developed and released as part of the PrefPaint study Bui et al. (2025). This model was trained on their custom dataset of nearly 51,000 human preference annotations. Unlike a general-purpose aesthetics model, it is specifically tailored to judge the quality of image inpainting, considering factors like structural rationality, local texture coherence, and overall aesthetic feeling. The reward

scores reported in our table are the normalized values from this model, averaged over the official test set, as done in the original paper.

## J PERSONALIZED IMAGE GENERATION EXPERIMENTAL DETAILS

This appendix provides a detailed overview of the experimental setup for evaluating Reinforcement Learning Guidance (RLG) on the task of personalized image generation, as presented in the main paper.

### J.1 TASK AND MODEL BACKGROUND

**Task Definition.** Personalized image generation aims to synthesize novel images of a specific subject provided through one or more reference images. The model is given a reference image containing the subject (e.g., a specific pet dog) and a text prompt (e.g., "a photo of [V] sleeping on a couch," where [V] is a placeholder for the subject). The primary goal is to generate an image that not only matches the prompt's description but also maintains high fidelity to the unique appearance and characteristics of the subject in the reference image.

**Model Selection.** Our experiment is designed to test if RLG can amplify the effects of a fine-grained, RL-based alignment process. We therefore select models based on the work of PatchDPO Huang et al. (2025).

- **Base Model ($\mathbf{v_{ref}}$):** We use the publicly available, pre-trained **IP-Adapter-Plus** Ye et al. (2023) model built on SDXL Podell et al. (2023). IP-Adapter is a powerful method for subject-driven generation that injects image features into the cross-attention layers of a diffusion model. We use it as our baseline because it represents a strong, general-purpose personalization model before any preference-based fine-tuning.
- **RL-aligned Model ($\mathbf{v_\theta}$):** We use the model fine-tuned from IP-Adapter-Plus using the **PatchDPO** algorithm. PatchDPO is a form of preference optimization that operates at a sub-image or "patch" level. During its training, generated images are compared against the reference image. Patches from the generated image that are consistent with the reference subject receive a positive reward, while inconsistent patches are penalized. This process, analogous to reinforcement learning with fine-grained rewards, tunes the model to be highly specialized in preserving subject fidelity.

### J.2 BENCHMARK AND EVALUATION METRICS

**Benchmark Dataset.** All evaluations are conducted on **DreamBench** Ruiz et al. (2023), the standard benchmark for personalized image generation. DreamBench consists of 30 unique subjects, each with a set of reference images and 80 corresponding text prompts. This benchmark is designed to test a model's ability to generate the subject in various contexts, poses, and interactions.

**Evaluation Metrics.** To quantitatively measure the fidelity of the generated images to the reference subject, we employ two standard, complementary metrics. For each prompt in DreamBench, we generate an image and compare it to the ground-truth reference images of the subject.

- **CLIP-I (Image Similarity):** This metric, introduced by the DreamBooth authors, measures the semantic similarity between the generated and reference images. It works by encoding both images into high-dimensional feature vectors using a pre-trained CLIP ViT-L/14 image encoder. The final score is the average cosine similarity between the embedding of the generated image and the embeddings of the reference images. A higher CLIP-I score indicates that the generated image is semantically and stylistically closer to the reference subject from the perspective of the CLIP model.
- **DINO (Structural Similarity):** This metric uses features extracted from a self-supervised ViT-S/16 DINO Caron et al. (2021) model. DINO is trained without labels and learns to capture rich information about object structure, texture, and shape. The metric is calculated as the average cosine similarity between the DINO features of the generated and reference

images. It is particularly effective at measuring the preservation of fine-grained details and the structural integrity of the subject, making it an excellent indicator of subject fidelity.

## K  DETAILED HUMAN PREFERENCE ALIGNMENT RESULTS

This section provides the complete quantitative results for the human preference alignment experiments, complementing the summary presented in Table 1 of the main paper. For each model, we present two tables: one detailing the absolute mean scores for Aesthetic Score, ImageReward, and PickScore across various RLG guidance scales ($w_{RL}$), and another showing the pairwise win rates against the base model ($w_{RL} = 0.0$) and the standard RL-finetuned model ($w_{RL} = 1.0$).

Table 7: Mean scores for the **SD3.5-M** model series on human preference metrics across various RLG scales ($w_{RL}$). The scale $w_{RL} = 0.0$ corresponds to the original **SD3.5-M** base model, while $w_{RL} = 1.0$ represents the model after GRPO finetuning, named **SD3.5M-FlowGRPO-PickScore**.

| $w_{RL}$ | Aesthetic Score ($\uparrow$) | ImageReward ($\uparrow$) | PickScore ($\uparrow$) |
|---|---|---|---|
| 0.0 | 5.97 | 0.99 | 21.75 |
| 1.0 | 6.45 | 1.40 | 23.29 |
| 1.2 | 6.48 | 1.41 | 23.36 |
| 1.4 | 6.54 | 1.40 | 23.48 |
| 1.6 | 6.57 | 1.40 | 23.53 |
| 1.8 | 6.60 | 1.41 | 23.56 |
| 2.0 | 6.62 | 1.40 | 23.57 |
| 2.2 | 6.64 | 1.39 | 23.58 |
| 2.4 | 6.66 | 1.39 | 23.58 |
| 2.6 | 6.68 | 1.37 | 23.59 |
| 2.8 | 6.68 | 1.36 | 23.56 |

Table 8: Win rates (%) for **SD3.5-M** model series at various RLG scales ($w_{RL}$) compared against the base ($w_{RL} = 0.0$) and GRPO ($w_{RL} = 1.0$) models.

| $w_{RL}$ | Win Rate vs. Base ($w_{RL} = 0.0$) | | | Win Rate vs. GRPO ($w_{RL} = 1.0$) | | |
|---|---|---|---|---|---|---|
| | Aesthetic | ImageReward | PickScore | Aesthetic | ImageReward | PickScore |
| 1.0 | 88.67 | 82.71 | 97.61 | - | - | - |
| 1.2 | 89.60 | 83.59 | 98.14 | 57.96 | 53.42 | 60.25 |
| 1.4 | 91.31 | 82.71 | 97.80 | 69.29 | 54.44 | 74.95 |
| 1.8 | 92.19 | 80.76 | 97.71 | 75.24 | 54.25 | 76.27 |
| 2.2 | 92.72 | 78.91 | 97.07 | 77.39 | 53.56 | 73.68 |
| 2.4 | 92.87 | 77.54 | 96.78 | 79.10 | 51.66 | 72.80 |

Table 9: Mean scores for **SDXL-base** model series on human preference metrics at various RLG scales ($w_{RL}$). The scale $w_{RL} = 0.0$ corresponds to the original **SDXL-base** base model, while $w_{RL} = 1.0$ represents the model after SPO finetuning, named **SPO-SDXL_4k-p_10ep**.

| $w_{RL}$ | Aesthetic Score ($\uparrow$) | ImageReward ($\uparrow$) | PickScore ($\uparrow$) |
|---|---|---|---|
| 0.0 | 6.10 | 0.72 | 21.66 |
| 1.0 | 6.42 | 1.12 | 22.69 |
| 1.2 | 6.45 | 1.13 | 22.71 |
| 1.4 | 6.48 | 1.14 | 22.71 |

| $w_{\mathrm{RL}}$ | Win Rate vs. Base ($w_{\mathrm{RL}} = 0.0$) | | | Win Rate vs. SPO ($w_{\mathrm{RL}} = 1.0$) | | |
|---|---|---|---|---|---|---|
| | Aesthetic | ImageReward | PickScore | Aesthetic | ImageReward | PickScore |
| 1.0 | 82.13 | 81.98 | 92.38 | - | - | - |
| 1.2 | 83.06 | 81.98 | 92.19 | 59.81 | 54.10 | 54.64 |
| 1.4 | 83.64 | 81.20 | 91.46 | 62.99 | 54.15 | 54.35 |

Table 10: Win rates (%) for **SDXL-base** model series at various RLG scales ($w_{\mathrm{RL}}$) compared against the base ($w_{\mathrm{RL}} = 0.0$) and SPO ($w_{\mathrm{RL}} = 1.0$) models.

Table 11: Mean scores for **SD1.5** model series on human preference metrics at various RLG scales ($w_{\mathrm{RL}}$). The scale $w_{\mathrm{RL}} = 0.0$ corresponds to the original **SD1.5** base model, while $w_{\mathrm{RL}} = 1.0$ represents the model after DPO finetuning, named **dpo-sd1.5-text2image-v1**.

| $w_{\mathrm{RL}}$ | Aesthetic Score ($\uparrow$) | ImageReward ($\uparrow$) | PickScore ($\uparrow$) |
|---|---|---|---|
| 0.0 (Base) | 5.51 | -0.02 | 20.03 |
| 1.0 (DPO) | 5.61 | 0.20 | 20.39 |
| 1.2 | 5.62 | 0.22 | 20.42 |
| 1.4 | 5.62 | 0.25 | 20.46 |
| 1.6 | 5.64 | 0.26 | 20.51 |
| 1.8 | 5.63 | 0.29 | 20.51 |
| 2.0 | 5.64 | 0.31 | 20.54 |
| 2.2 | 5.64 | 0.31 | 20.55 |
| 2.4 | 5.64 | 0.32 | 20.56 |

Table 12: Win rates (%) for **SD1.5** at various RLG scales ($w_{\mathrm{RL}}$) compared against the base ($w_{\mathrm{RL}} = 0.0$) and standard DPO ($w_{\mathrm{RL}} = 1.0$) models.

| $w_{\mathrm{RL}}$ | Win Rate vs. Base ($w_{\mathrm{RL}} = 0.0$) | | | Win Rate vs. DPO ($w_{\mathrm{RL}} = 1.0$) | | |
|---|---|---|---|---|---|---|
| | Aesthetic | ImageReward | PickScore | Aesthetic | ImageReward | PickScore |
| 1.0 | 61.52 | 63.33 | 72.66 | - | - | - |
| 1.4 | 64.11 | 63.82 | 75.34 | 53.56 | 53.47 | 57.86 |
| 1.8 | 63.92 | 66.36 | 76.27 | 55.71 | 56.69 | 61.43 |
| 2.2 | 64.55 | 66.46 | 76.22 | 56.40 | 56.64 | 61.72 |
| 2.4 | 64.21 | 66.31 | 76.61 | 56.25 | 57.08 | 61.23 |

# L SELECTED IMAGE GENERATED

## L.1 AESTHETIC IMAGES GENERATED

## L.2 OCR IMAGES GENERATED

## L.3 COMPRESSIBILITY AND INCOMPRESSIBILITY IMAGES GENERATED

## L.4 GENEVAL IMAGES GENERATED

## L.5 INPAINTING IMAGES GENERATED

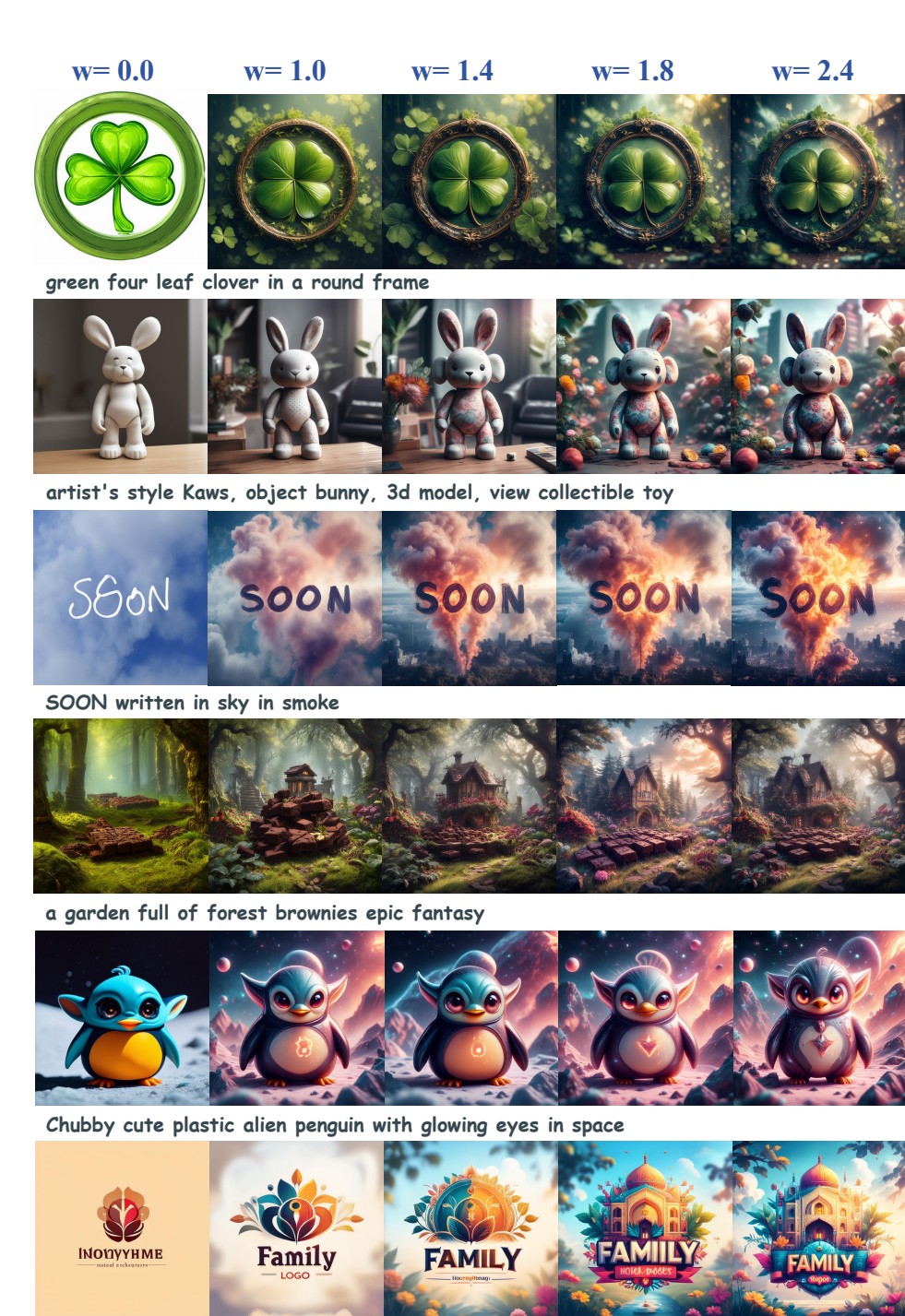

Figure 5: Selected qualitative results for the human preference task. Images are generated from SD3.5 trained with GRPO, with different RLG scales.

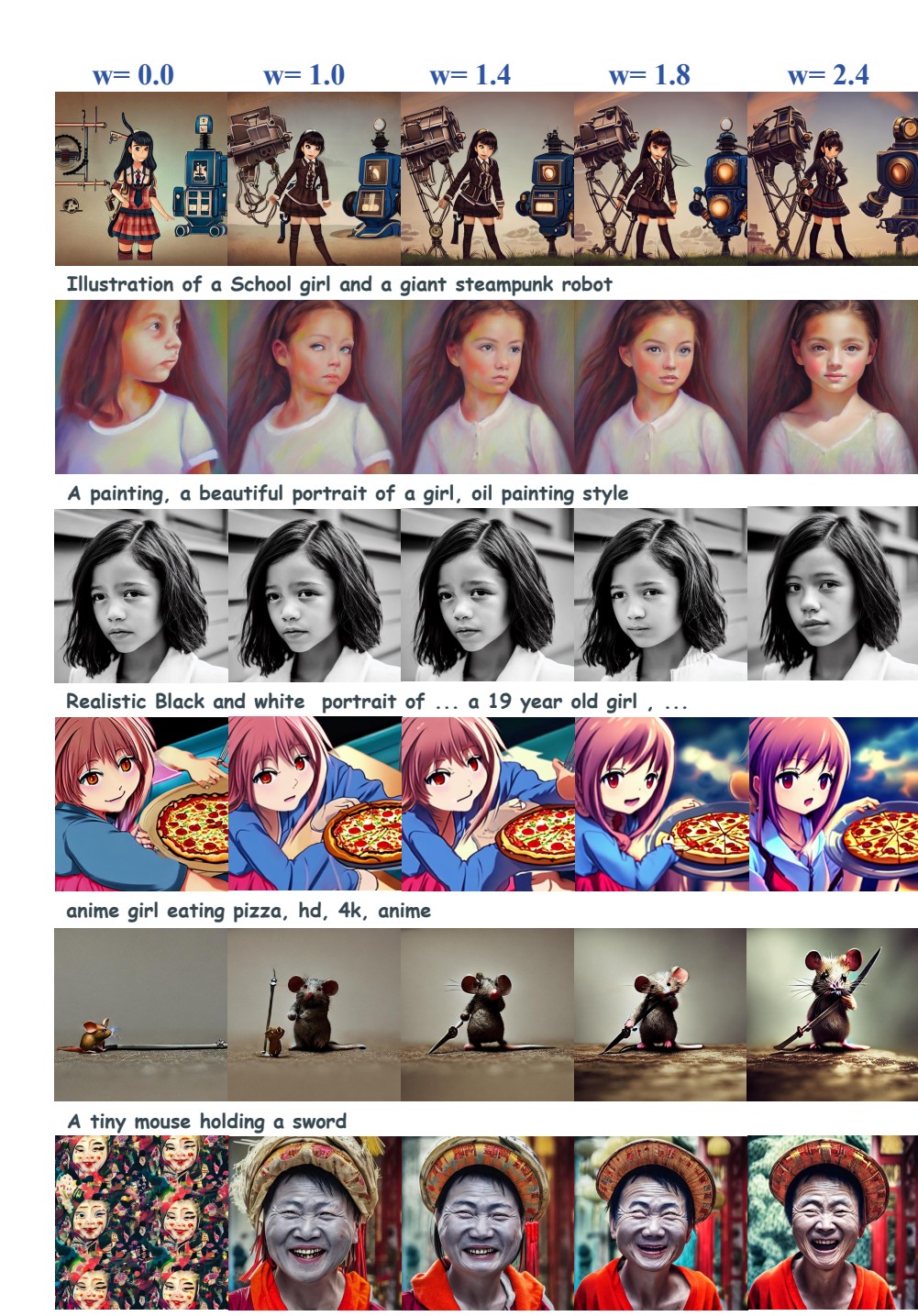

Figure 6: Selected qualitative results for the human preference task. Images are generated from SD1.5 trained with DPO, with different RLG scales.

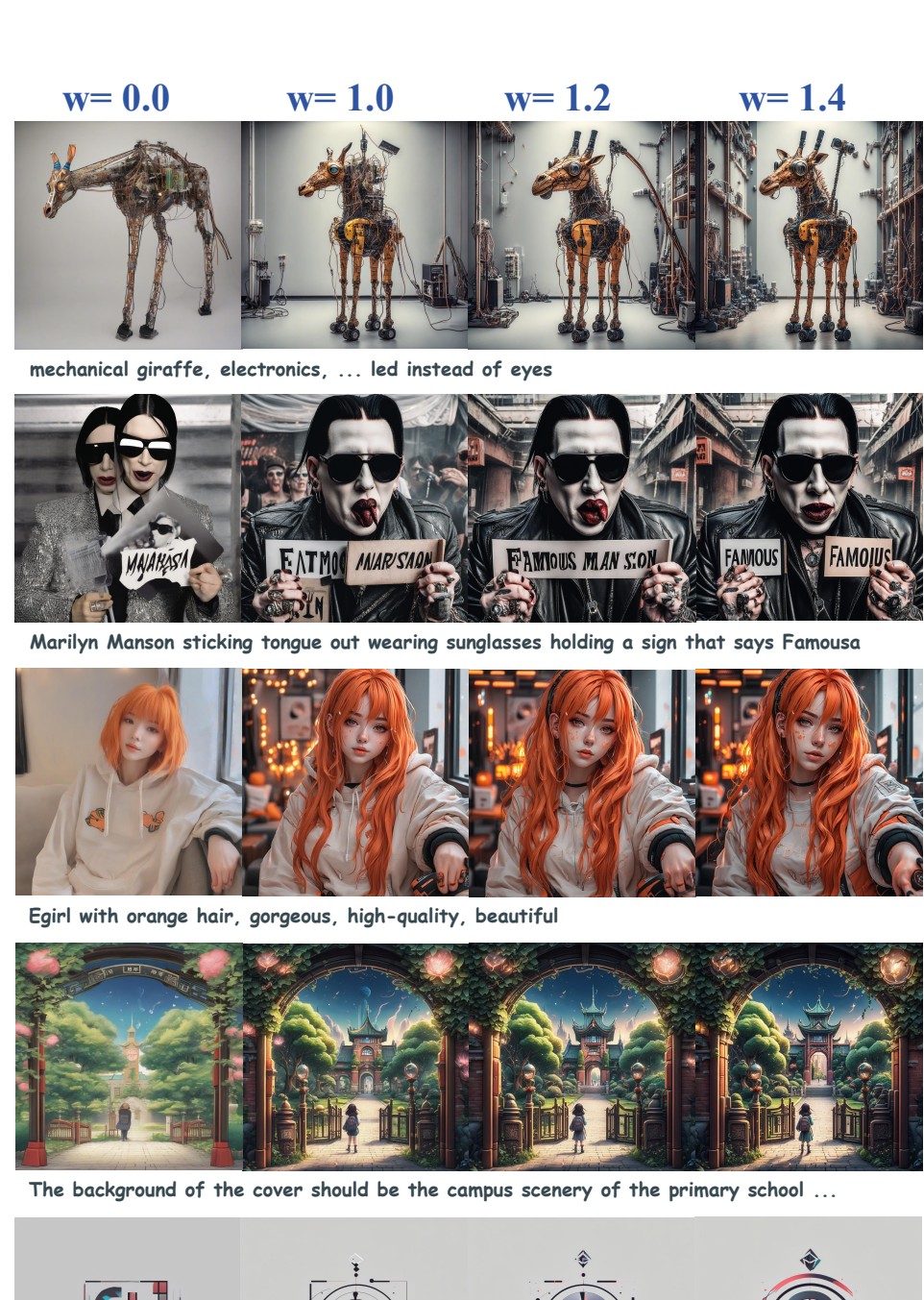

Figure 7: Selected qualitative results for the human preference task. Images are generated from SDXL trained with SPO, with different RLG scales.

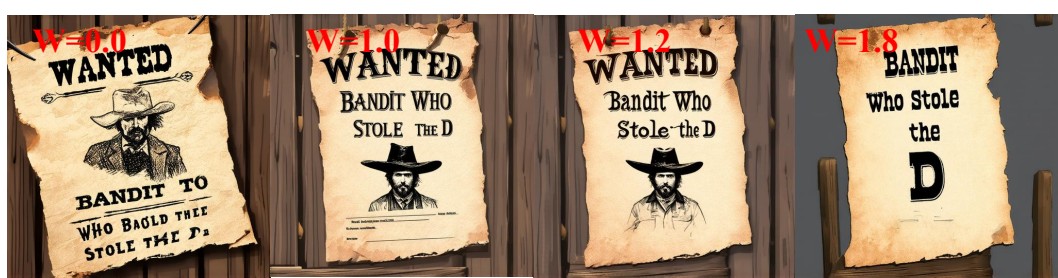

" Bandit Who Stole the D "

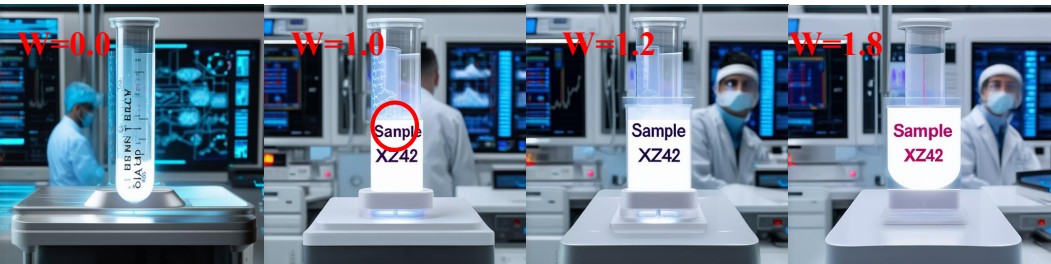

" Sample XZ42 "

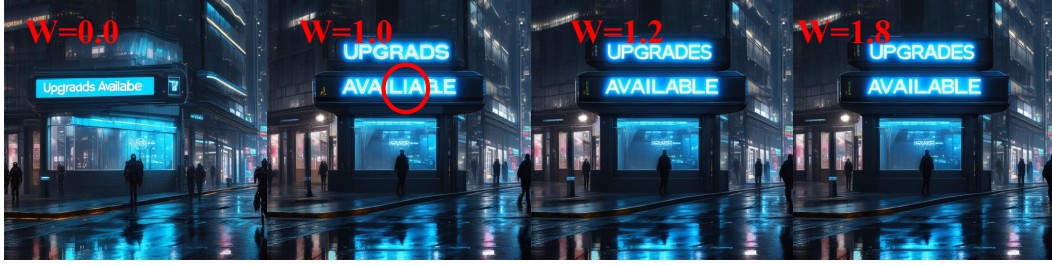

" Upgrades Available "

Figure 8: Selected qualitative results for the visual text rendering task. As can be seen, the standard RL-finetuned model ($w = 1.0$) still produces some errors in the generated text. By applying RLG with a higher guidance scale ($w > 1.0$), the model correctly renders the specified text without any loss in image quality. This illustrates how RLG effectively enhances the model's ability to adhere to precise instructions.

w= 0.0 w= 1.0 w= 1.6 w= 2.4

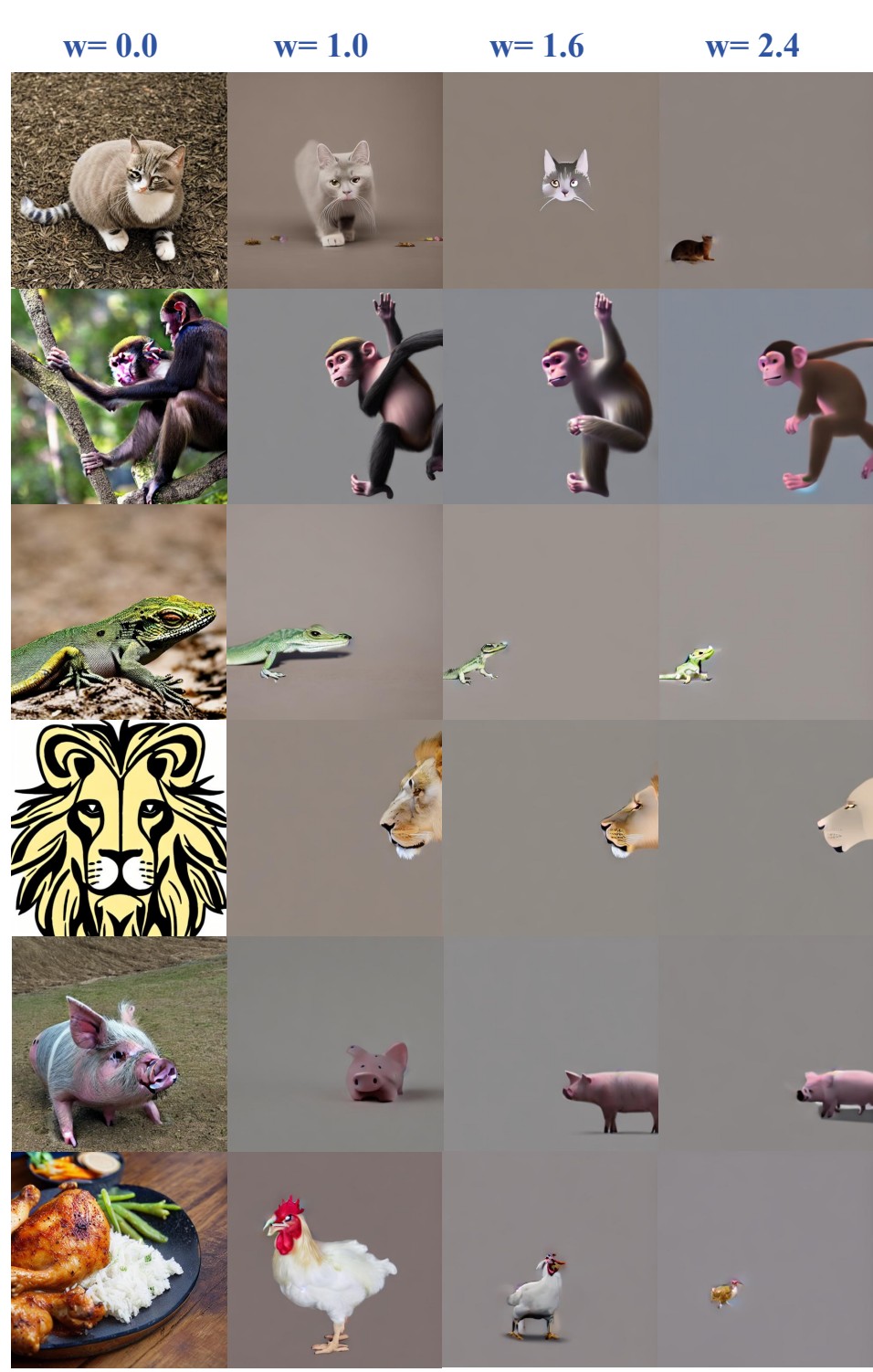

Figure 9: Selected qualitative results for the image compressibility task.

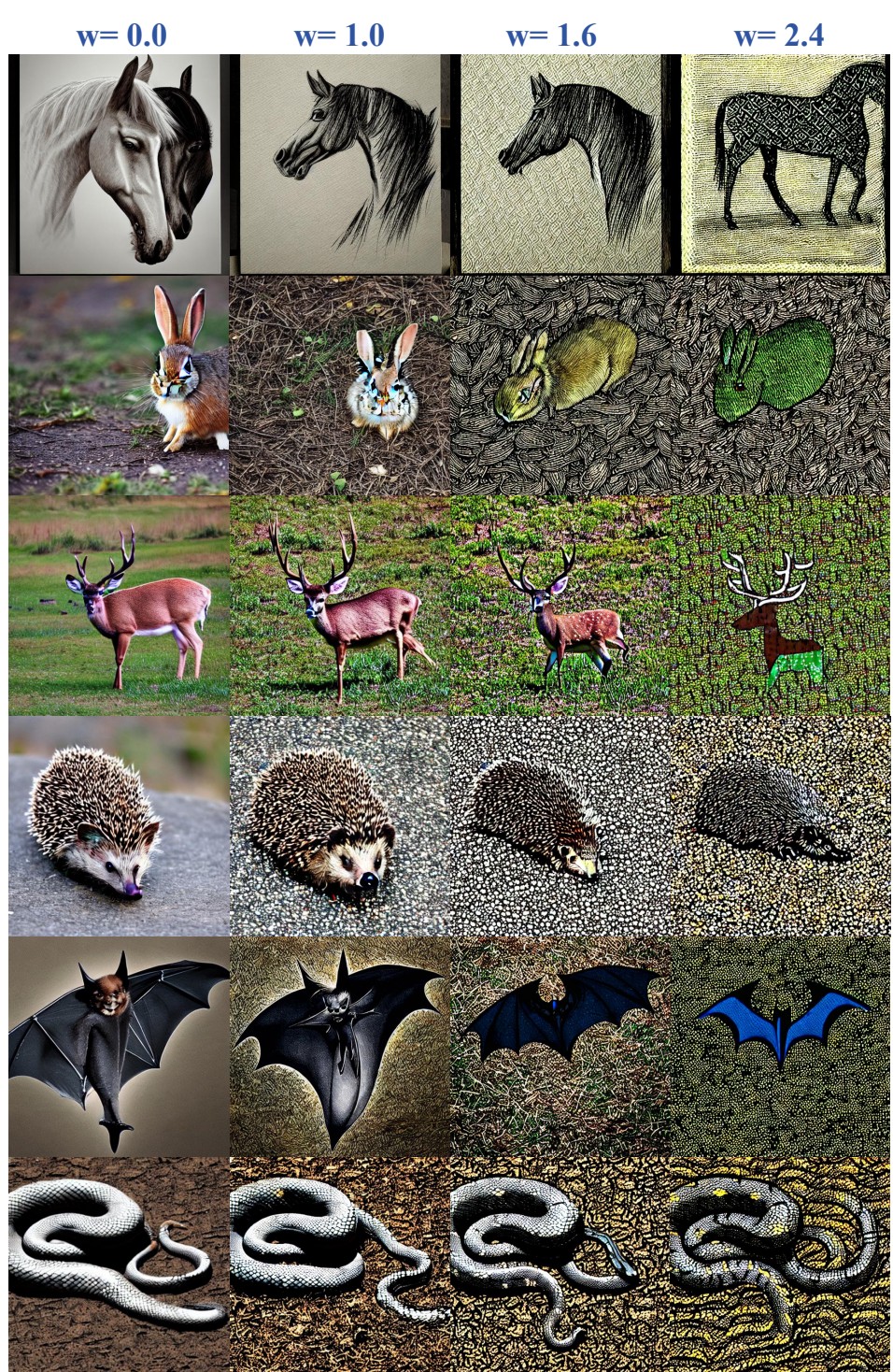

Figure 10: Selected qualitative results for the image compressibility task.

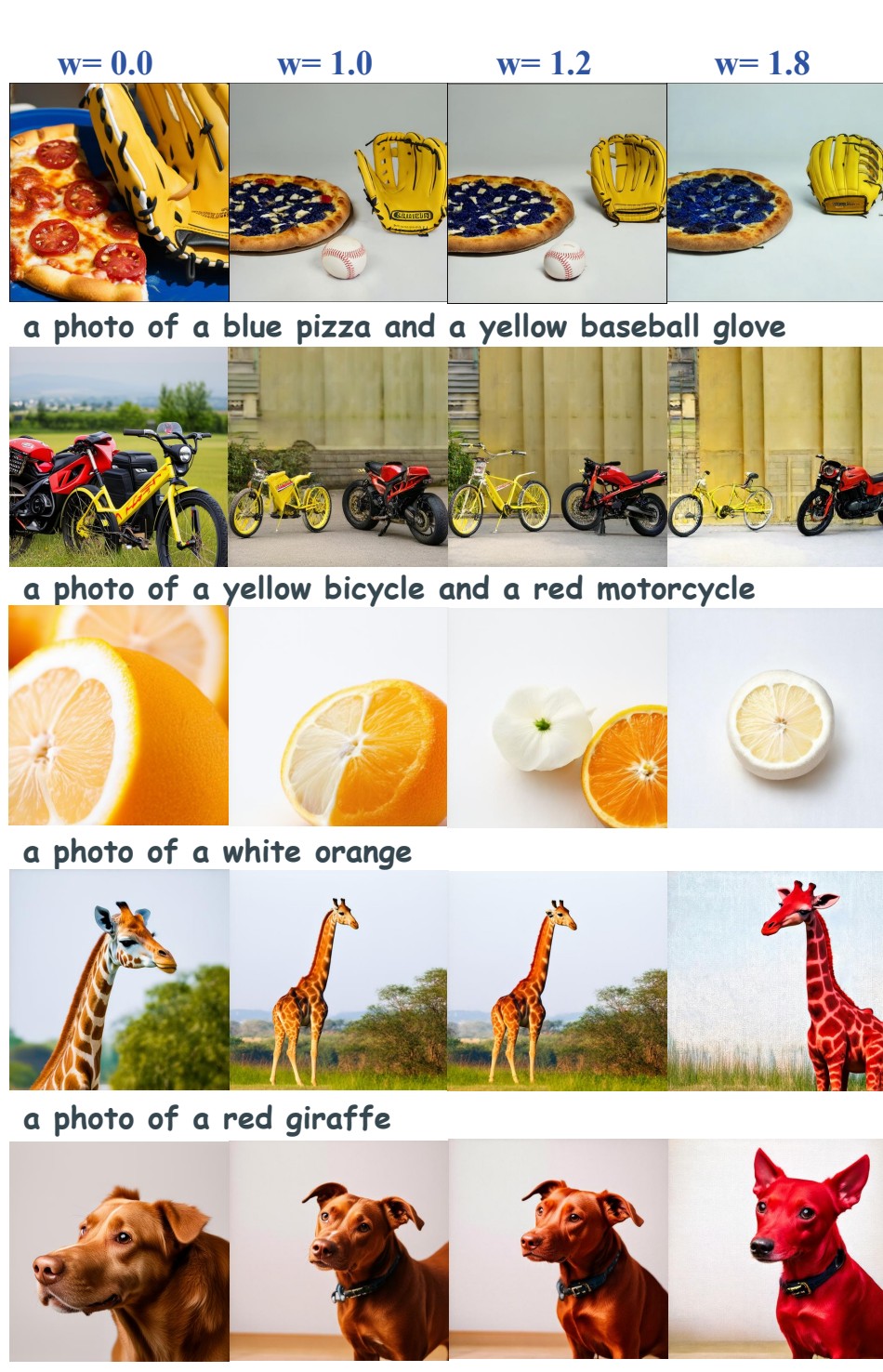

Figure 11: Selected qualitative results for the compositional image generation task.

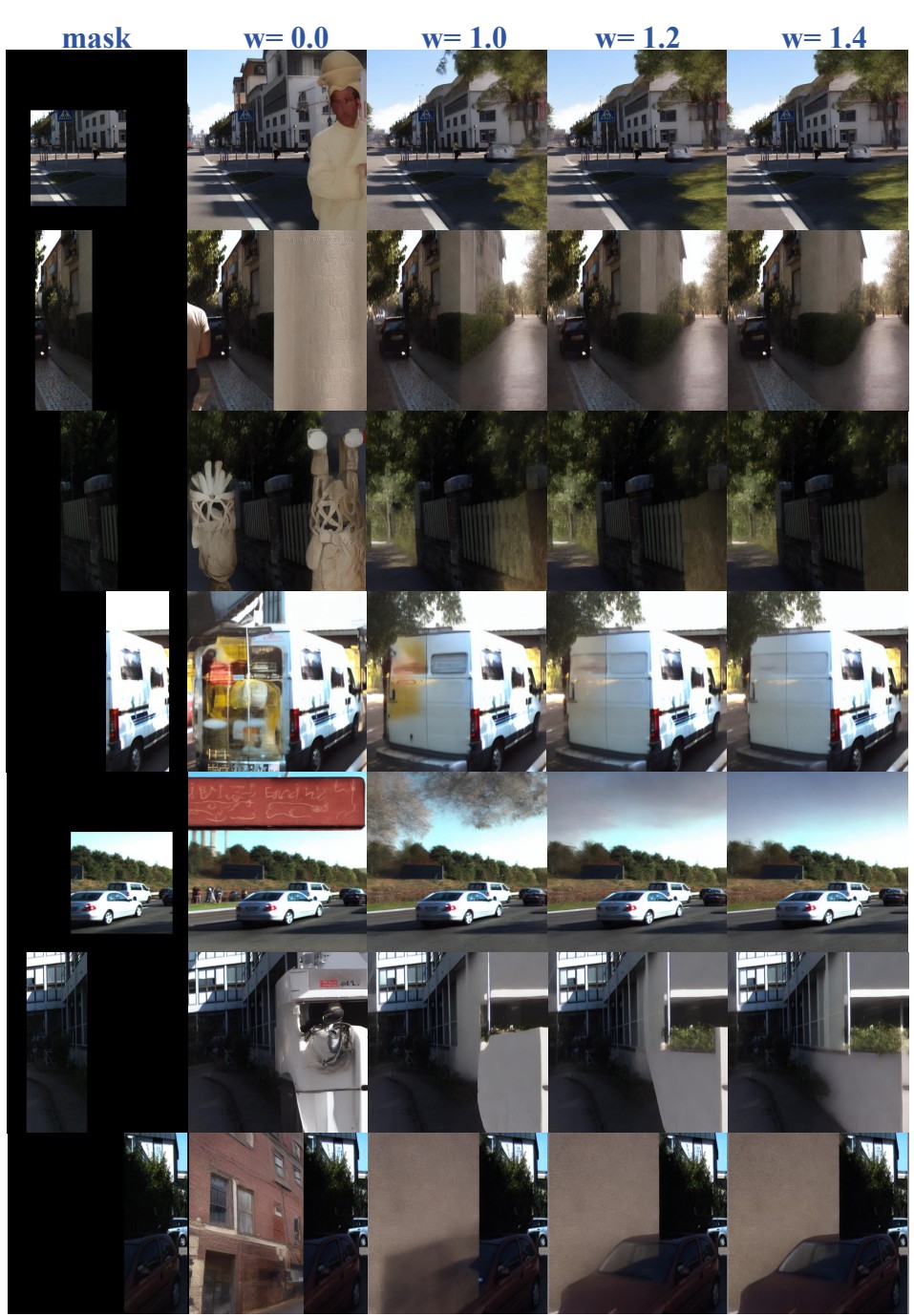

Figure 12: Selected qualitative results for the image inpainting task.

