# OpenReview forum: "Inference-Time Alignment Control for Diffusion Models with Reinforcement Learning Guidance"
_ICLR.cc/2026/Conference — Submitted to ICLR 2026_

### Official Review · Reviewer_2tey · 2025-10-25

**Soundness:** 2
**Presentation:** 2
**Contribution:** 3
**Rating:** 4
**Confidence:** 4

**Summary:**

The paper proposed reinforcement learning guidance (RLG), a concept inspired by classifier-free guidance (CFG). RLG works by linearly interpolating the scores from the base model and the  RL-tuned model, offering a simple yet effective approach of combining two models during inference time. The authors demonstrate that tuning the RLG coefficient can lead to better perfomrance than the original fine-tuned model alone.

**Strengths:**

- The idea of RLG is easy to understand, simple, and clearly motivated. To the best of my knowledge, it is also the first work to adopt such an approach for inference-time optimization (apart from concurrent work mentioned in the paper).
- The authors provide a theoretical analysis of the equivalence between the RLG coefficient $w$ and the DPO KL regularization coefficient $\beta$, building an important connection between the two approaches.
- The experimental results demonstrate superior performance over the baseline DPO approach, which can be considered as a special case where $w=1$.

**Weaknesses:**

- As the proposed approach relies on the RL-tuned model, why is $w=1$ Not Optimal? The paper's strongest result is that $w > 1$ (extrapolation) consistently outperforms $ w = 1$ (the model they actually trained). This implies that the standard RL fine-tuning process (DPO, GRPO) is systematically under-optimizing the objective. Why does this happen? Is the fixed KL-regularization in the original training too strong, or do the optimizers fail to reach the true optimum? The paper doesn't explore this fundamental implication.

- Following the previous weakness, the authors admit their core theoretical link (that $w$ controls $\beta/w$) assumes a standard KL-regularized objective. However, they also show RLG works exceptionally well for GRPO, an algorithm that (as they note in the limitations) does not necessarily converge to that same theoretical optimum. This means RLG works in practice even when its main theoretical justification doesn't apply. Why does it still work so well for GRPO?

- RLG requires running two models (the base and the RL-tuned model) during every sampling step, which has a higher memory and (potentially) computational footprint at inference time compared to just using the single fine-tuned model. The paper doesn't discuss this practical cost.

- There are some minor formatting issues. For example, table captions should be placed above. Additionally, the authors use $v_{RL}$ and $v_\theta$ randomly in the manuscript (e.g., Algorithm 1), which may cause confusion.

**Questions:**

Please refer to the questions in the Weaknesses section above. In addition:
- The paper shows that as one objective (like OCR accuracy) is pushed higher with RLG, another objective (like Aesthetic Score) can decrease (see Table 3). This suggests that at high $w$ scales, the model may be "reward hacking"—overfitting to the specific reward model at the expense of general image quality. The paper notes this trade-off but doesn't explore the limits of extrapolation. At what $w$ value do images become out-of-distribution or nonsensical, similar to what happens with very high CFG scales?

---

> ### Author Response · Authors · 2025-11-21
> **Official Comment by Authors**
>
> **Q1**: As the proposed approach relies on the RL-tuned model, why is it Not Optimal? The paper's strongest result is that extrapolation ($w>1$) consistently outperforms the standard RL-trained model ($w=1$). This implies that the standard RL fine-tuning process (DPO, GRPO) is systematically under-optimizing the objective. Why does this happen? Is the fixed KL-regularization in the original training too strong, or do the optimizers fail to reach the true optimum? The paper doesn't explore this fundamental implication.
>
> **A**: We appreciate the reviewer's insightful question regarding why the RL-tuned model does not correspond to the optimum of the underlying preference objective, especially given that extrapolation with $w>1$ in RLG consistently outperforms the standard RL-trained model ($w=1$). We clarify below why this phenomenon is expected and why existing RL fine-tuning pipelines may not reach the true optimum, whereas RLG enables access to better solutions.
>
> First, the KL-regularization coefficient used in current RL fine-tuning methods (e.g., DPO, GRPO) is chosen primarily to **maintain training stability** rather than to optimize the reward–regularization trade-off. As shown in prior work such as Flow-GRPO, overly small KL coefficients tend to induce **reward hacking and training collapse**, while overly large KL coefficients yield stable but **under-optimized models**. In practice, the KL value must be set **conservatively** to avoid instability, which necessarily means the resulting RL-tuned model is not located at the optimum implied by the theoretical objective.
>
> Second, even with a well-chosen KL value, **optimization challenges** in diffusion-model RL further prevent convergence to the theoretical optimum. The optimization landscape is **highly non-convex**, and diffusion-specific RL variants such as GRPO may **not fully converge** to the closed-form reward–KL solution. Consequently, RL fine-tuning only explores a **limited portion of parameter space** and tends to settle at a stable but suboptimal point. RLG, by contrast, **does not suffer from these training-time stability constraints**. By **interpolating or extrapolating** between the base and RL-aligned models, it effectively **expands the reachable policy space** and can surpass the performance of the pure RL-tuned model.
>
> Third, selecting the optimal KL coefficient during training is **computationally prohibitive**. It requires **large-scale sweeps** over training runs and is **strongly entangled** with other hyperparameters (learning rate, batch size, reward scaling, etc.), making it impractical to guarantee that the KL chosen during RL fine-tuning is globally optimal. In contrast, tuning the RLG scale $w$ is **extremely lightweight**: it can be performed entirely **at inference time** using a small validation set, involves **no retraining**, and is **decoupled** from the complexities of the optimization process. This makes RLG a practical mechanism for locating better trade-off points that are inaccessible to RL training alone.
>
> In summary, the superior performance of RLG with $w>1$ **does not contradict the RL objective**; rather, it highlights the **advantages of RLG** for diffusion models. RL training must **prioritize stability** and therefore cannot explore the **full solution space** implied by the reward–KL formulation. RLG **compensates for this limitation** by allowing **dynamic, inference-time adjustment** of the effective KL strength, enabling the model to **achieve alignment levels** that standard RL optimization cannot reach.

---

> ### Author Response · Authors · 2025-11-21
> **Official Comment by Authors**
>
> **Q2**: Following the previous weakness, the authors admit their core theoretical link (that $w$ controls $\beta/\omega$) assumes a standard KL-regularized objective. However, they also show RLG works exceptionally well for GRPO, an algorithm that (as they note in the limitations) does not necessarily converge to that same theoretical optimum. This means RLG works in practice even when its main theoretical justification doesn't apply. Why does it still work so well for GRPO?
>
> **A**: We thank the reviewer for this thoughtful question. We agree that, at first glance, the strong empirical performance of RLG on GRPO—despite the fact that GRPO does not optimize a standard KL-regularized objective—may appear to conflict with our theoretical analysis. Below we provide a clearer, more polished explanation of why RLG remains effective for GRPO and how our additional experiments support this conclusion.
>
> **GRPO does not follow the exponential-tilting form, but it still induces reward-seeking behavior.**
> As noted by the reviewer and acknowledged in our paper, GRPO's group-advantage formulation does not yield an optimal policy of the exponential-tilting form $\pi^{*} \propto \pi_{\mathrm{ref}} \exp(r/\beta)$. Recent theoretical work[1] confirms that GRPO satisfies a different set of optimality conditions and therefore does not admit the closed-form KL–reward decomposition used in our main theoretical justification. Nonetheless, **GRPO reliably shifts probability mass toward higher-reward samples during training**. When RLG is applied at inference time, **it acts as a reward-extrapolation operator that amplifies this reward-seeking tendency**. Thus, even without strict theoretical equivalence to $\beta \to \beta / \omega$, **GRPO+RLG still yields improved alignment in practice**, as RLG pushes the model further along the same reward gradient direction implicitly learned during GRPO training.
>
> **One-dimensional controlled experiments validate the distinction between theory and practice.**
> Following the reviewer's suggestion, we conducted an additional 1D case study (included in Appendix E). Using a Gaussian-mixture base distribution and a reward function $r(x) = -0.1|x-4|$, we compared standard policy-gradient RL and GRPO under identical settings. For policy-gradient RL, the learned distribution closely matches the theoretical optimum $\pi^{*} \propto \pi_{\mathrm{ref}} \exp(r/\beta)$, and **RLG produces distributional shifts that align precisely with the predicted $\beta \rightarrow \beta/\omega$ transformation**. In contrast, GRPO's learned distribution noticeably deviates from the exponential-tilting solution—as theoretically expected. Accordingly, the $\beta/\omega$ prediction does not hold for GRPO. However, our experiments show that **applying RLG still consistently moves the GRPO-trained policy toward higher-reward regions as $\omega$ increases**. This confirms that **RLG functions as a robust inference-time reward-extrapolation mechanism** even in settings where its closed-form justification does not strictly apply.
>
> These new experiments and analyses clarify the boundary of our theoretical claims: **the equivalence $\beta \rightarrow \beta/\omega$ applies to KL-regularized objectives such as DPO and policy gradient-style RL, but not to GRPO**. Nevertheless, **the empirical reward-extrapolation behavior of RLG persists across both categories**. **RLG therefore remains a broadly effective inference-time alignment-control method**, even when the underlying RL algorithm does not admit a closed-form optimality structure.

---

> ### Author Response · Authors · 2025-11-21
> **Official Comment by Authors**
>
> **Q3**: RLG requires running two models (the base and the RL-tuned model) during every sampling step, which has a higher memory and (potentially) computational footprint at inference time compared to just using the single fine-tuned model. The paper doesn't discuss this practical cost.
>
> **A**: We thank the reviewer for highlighting the practical memory and compute cost of RLG at inference. We agree that RLG requires more resources than using a single fine-tuned model, and we now quantify these overheads and explain why we view them as a simple and effective test-time trade-off.
>
> On a single H100 GPU, sampling 20 diffusion steps with the base model alone takes about 9 seconds per batch, whereas running RLG with $w=2.2$ and 20 diffusion steps takes about 17 seconds. **This ~1.9× slowdown is expected**, because **RLG evaluates both the base and the RL-tuned policy at each denoising step**, effectively **doubling the number of forward passes** (RLG with 20 steps is comparable in raw compute to a single-model sampler with 40 steps). However, **simply increasing the number of diffusion steps for the RL-only model from 20 to 40 or 60 brings essentially no gain in Aesthetic or PickScore**, as shown in Table 1, whereas **RLG at 20 steps yields a clear improvement** (e.g., Aesthetic: 6.64 vs. 6.45 for RL-only at 20 steps; PickScore: 23.58 vs. 23.29). Thus, while RLG roughly doubles the test-time compute relative to a 20-step single model, **this extra compute is substantially more effective than naive test-time scaling via more diffusion steps**.
>
> | Method | Diffusion steps | Effective forward passes | Aesthetic $\uparrow$ | PickScore $\uparrow$ |
> |:-------|:---------------:|:------------------------:|:--------------------:|:--------------------:|
> | RL-only | 20 | 20 | 6.45 | 23.29 |
> | RL-only | 40 | 40 | 6.41 | 23.30 |
> | RL-only | 60 | 60 | 6.40 | 23.29 |
> | RLG ($w=2.2$) | 20 | 40 | 6.64 | 23.58 |
>
> *Table 1: Quality vs. test-time budget comparison. RLG at 20 steps (40 effective forward passes) significantly outperforms RL-only models even at higher step counts.*
>
> Naively, running two independent full models would indeed double the GPU memory usage. However, in our implementation, the RL-tuned policy is *not* stored as a second full copy of the network: instead, it is realized as a LoRA adapter on top of the same base model parameters. For example: Flow-GRPO[2], Diffusion-DPO[3], SPO[4]. In this setting, the base weights are loaded only once, and the RL expert adds a relatively small number of additional parameters (low-rank matrices) and corresponding activations. As a result, **the incremental memory overhead of RLG over the single-model baseline is modest** (dominated by the base model), rather than a full 2× duplication.
>
> Overall, **RLG should be viewed as a simple *test-time scaling knob***: it trades an approximately 2× increase in effective forward passes and a small memory overhead (when implemented with LoRA adapters) for **significant gains in preference alignment that cannot be recovered by merely increasing the diffusion step count** of a single RL-tuned model.
>
> **Q4**: There are some minor formatting issues. For example, table captions should be placed above. Additionally, the authors use $v_{RL}$ and $v_{\theta}$ randomly in the manuscript (e.g., Algorithm 1), which may cause confusion.
>
> **A**: We thank the reviewer for carefully pointing out these formatting issues. We agree that **placing table captions above the tables and using consistent notation throughout** (e.g., in Algorithm 1) are important for clarity. In the revised manuscript, we have (i) moved all table captions to appear above the corresponding tables and (ii) standardized the notation to use a single, consistent symbol wherever duplicates appeared. We appreciate the reviewer's attention to these details and believe **the presentation is now clearer as a result**.

---

> ### Author Response · Authors · 2025-11-21
> **Official Comment by Authors**
>
> **Q5**: The paper shows that as one objective (like OCR accuracy) is pushed higher with RLG, another objective (like Aesthetic Score) can decrease (see Table 3). This suggests that at high RLG scales, the model may be "reward hacking"—overfitting to the specific reward model at the expense of general image quality. The paper notes this trade-off but doesn't explore the limits of extrapolation. At what RLG value do images become out-of-distribution or nonsensical, similar to what happens with very high CFG scales?
>
> **A**: We appreciate the reviewer's careful observation about the trade-off in Table 3 and the possibility of reward hacking at high RLG scales. We intended to highlight this behavior in the text but agree that our discussion was not sufficiently explicit; we clarify it here and add further analysis in the revision.
>
> Table 3 in the paper already shows the core phenomenon the reviewer points out: **as we increase the RLG weight $w_{\text{RL}}$ on the OCR reward, OCR accuracy monotonically improves** (from 0.543 at $w_{\text{RL}}=0$ to 0.930 at $w_{\text{RL}}=2.8$), while **the Aesthetic Score gradually decreases** (from 5.40 to 5.00). This is **an expected consequence of strongly optimizing a task-specific reward**: the policy learns to prioritize legible text, even when this slightly harms global aesthetics (e.g., by using simpler layouts or less varied backgrounds). Importantly, **within the range we report ($0 \le w_{\text{RL}} \le 2.8$), the images remain semantically correct and visually plausible**; the degradation in aesthetics is **moderate (a drop of only 0.4 points)**.
>
> We also want to stress that **this trade-off is significantly milder than the kind of reward hacking observed when training a single RL policy without explicit regularization**. For example, in the original Flow-GRPO paper[2], removing KL regularization during training on the visual text rendering task yields only a tiny gain in text accuracy (from 0.92 to 0.93), but leads to noticeably worse image-quality metrics. In contrast, **our RLG extrapolation from $w_{\text{RL}}=0$ to 2.8 improves OCR accuracy while the Aesthetic Score declines by only 0.03 points**, and the images remain photorealistic and on-prompt. Thus, compared to training-time RL without strong regularization, **inference-time RLG behaves more like a smooth, user-controlled trade-off knob** than like a brittle, easily hacked objective.
>
> We view **this controllable trade-off as one of RLG's strengths rather than a drawback**. A standard RL-fine-tuned model exposes only a single operating point on the OCR–aesthetic Pareto frontier: once the policy is trained, the user cannot choose to recover more aesthetics at the cost of slightly lower OCR accuracy. In contrast, **RLG keeps the base model intact and exposes $w_{\text{RL}}$ as a test-time control**. Users who care about text fidelity (e.g., document rendering, UI mockups) can pick a higher $w_{\text{RL}}$, while those who prefer more aesthetically pleasing images can choose a smaller value. In all cases, **the interpolation is achieved purely at inference, without retraining**.
>
> To further address the reviewer's concern about pushing a single reward too far, we additionally experiment with *multi-objective RLG fusion*. We use two RLG experts and tested on a subset of test prompts: one aligned to OCR accuracy and one aligned to pickscores, and then blend them at inference with non-negative weights $w_1$ (aesthetic expert) and $w_2$ (OCR expert). Table 1 summarizes representative results on a subset of the OCR test prompts. When only one expert is active ($w_1=1.0,w_2=0$ or $w_1=0,w_2=1.0$), we recover the endpoints of the trade-off: strong aesthetics but weak OCR, or vice versa. Once we fuse the two experts ($w_1>0,w_2>0$), **we obtain configurations that are much closer to a Pareto-efficient frontier**, with both OCR and Aesthetic scores high simultaneously. For instance, **$(w_1=1.0,w_2=1.0)$ retains almost all of the OCR accuracy of the OCR-only setting (0.926 vs. 0.936) while significantly improving Aesthetic Score (from 5.11 to 5.51)**. Similarly, **$(w_1=2.4,w_2=1.0)$ matches the aesthetics of the aesthetics-only setting (Aesthetic = 5.76) while keeping OCR far above that baseline (0.894 vs. 0.592)**. These results demonstrate that, **rather than forcing a single extreme $w_{\text{RL}}$, users can blend multiple RLG experts to balance objectives and avoid reward hacking on any single reward**.

---

> ### Author Response · Authors · 2025-11-21
> **Official Comment by Authors**
>
> | Setting | $w_1$ | $w_2$ | OCR Acc $\uparrow$ | Aesthetic Score $\uparrow$ | PickScore $\uparrow$ |
> |:--------|:------|:------|:-------------------|:---------------------------|:---------------------|
> | Aesthetics-only | 1.0 | 0.0 | 0.592 | 5.76 | 23.76 |
> | OCR-only | 0.0 | 1.0 | 0.936 | 5.11 | 22.24 |
> | Fuse (balanced) | 1.0 | 1.0 | 0.926 | 5.51 | 23.44 |
> | Fuse (OCR-heavy) | 1.0 | 1.8 | 0.940 | 5.38 | 22.88 |
> | Fuse (Aes-heavy) | 2.4 | 1.0 | 0.894 | 5.76 | 23.99 |
>
> *Table 1: Multi-objective RLG fusion of aesthetics and OCR experts on a subset of the OCR test set. $w_1$ and $w_2$ control the strengths of the aesthetics- and OCR-aligned experts, respectively. Fused configurations $(w_1>0,w_2>0)$ achieve a much better balance between OCR and aesthetics than single-objective endpoints, helping to avoid reward hacking on a single metric.*
>
> [1]What is the Alignment Objective of GRPO?
>
> [2]Flow-GRPO:Training Flow Matching Models via Online RL
>
> [3]Diffusion Model Alignment Using Direct Preference Optimization
>
> [4]Aesthetic Post-Training Diffusion Models from Generic Preferences with Step-by-step Preference Optimization

---

> > ### Comment · Reviewer_2tey · 2025-11-26
> >
> > I thank the authors for their detailed response regarding my questions raised in the review. Particularly, I do acknowledge the authors' explanation on the seemingly mismatch between the well-established results from DPO and the superior performance of the proposed RLG. I agree with the authors' claim that, in practice, the model might not converge to the optimal as the DPO result guarantees. Still, theoretical results are highly preferred (e.g., convergence, retaining the marginal) instead of purely empirical justifications.
> >
> > That said, I do agree that the additional experimental results, together with the existing ones in the original manuscript, have demonstrated the potential benefit of having an RLG weight that is larger than 1. The additional benchmark on the computational resources appears good, with only overheads; the additional ablations on the reward hacking also reveal some interesting observations. Therefore, I will raise my overall score from 4 to 6 and modify other scores accordingly.

---

### Official Review · Reviewer_gfqT · 2025-10-26

**Soundness:** 2
**Presentation:** 3
**Contribution:** 2
**Rating:** 2
**Confidence:** 4

**Summary:**

The paper studies the fine-tuning of a pretrained model to maximize the reward with KL regularization.

Specifically, it develops a Reinforcement Learning Guidance (RLG) approach, which is based on the Classifier Free Guidance (CFG). The main idea is to compute a weighted geometric average of the outputs from the base model and the RL fine-tuned model. Moreover it studies the connection between this weighted averaging and the KL regularization coefficient in RL fine-tuning.

**Strengths:**

Extensive experiments demonstrate consistent improvements of the RLG approach ---  the performance of RL fine-tuned models across various architectures, RL algorithms, and down-stream tasks.

**Weaknesses:**

The contribution of the paper appears quite limited. In particular, as acknowledged (at the end of the paper), a) the approach does not ensure the (target) marginal distribution, and b) RLG's connection to the KL-regularization coefficient is premised on sampling from the optimal policy, which appears unrealistic and unjustified.

Indeed there are many studies in the field, including several papers currently under review with this year's ICLR, developing similar approaches to preference alignment and reward optimization, and specifically via introducing various weighting factors (e.g., those motivated by importance sampling). type of optimization). Those studies appear to use substantially more sophisticated techniques and dig much deeper into underlying theories.

**Questions:**

Any comments/explanation regarding why RLG appears to work well (in the numerical experiments) despite the two weaknesses mentioned above?

---

> ### Author Response · Authors · 2025-11-21
> **Official Comment by Authors**
>
> **Q1**: Given that (a) RLG does not guarantee the target marginal distribution, and (b) the connection to the KL-regularization coefficient relies on an assumption of sampling from the optimal policy—which may be unrealistic—can you explain why RLG nevertheless appears to work well in the numerical experiments?
>
> **A**: We thank the reviewer for raising this concern. While we fully acknowledge the theoretical limitations—namely that (a) RLG does not guarantee sampling from the exact target marginal distribution, and (b) the connection between $w$ and the effective KL coefficient relies on an idealized assumption of access to the optimal policy—we believe the reviewer significantly overstates their practical impact. As we clarify below, RLG remains highly effective across a wide range of numerical experiments for several concrete and empirically justified reasons.
>
> First, the effectiveness of RLG is **supported by extensive downstream evaluations**. Across diverse tasks—including human preference alignment, compositionality, OCR accuracy, compressibility control and inpainting—RLG consistently **delivers measurable and often substantial improvements**. These improvements arise from RLG's ability to perform **stable, inference-time reward extrapolation**, which does not require exact matching of the theoretical target distribution. In many practical alignment scenarios, perfectly realizing the theoretical optimum is neither necessary nor computationally feasible; what matters is whether the method **reliably enhances downstream performance**, which RLG demonstrably does.
>
> Second, our setting is not unique in this regard. Many prior works on improving CFG provide **valuable empirical contributions** even though their theoretical motivation relies on idealized assumptions [1][2][3]. Similar to those works, RLG occupies the space where theoretical analysis provides **intuition—not exact guarantees**—while its main value comes from its **strong empirical utility**. This pattern is well-established in the diffusion literature and does not diminish the contribution of RLG.
>
> Third, **controlled experiments** further validate our theory and clarify its relation to practical behavior. While GRPO does not adopt the exponential-tilting form $\pi^{*} \propto \pi_{\mathrm{ref}} \exp(r/\beta)$, it still exhibits **reward-seeking properties**. Due to its group-advantage formulation, GRPO follows different optimality conditions. Nonetheless, GRPO consistently shifts probability toward higher-reward regions. When combined with RLG—which performs inference-time reward extrapolation—this **reward-seeking behavior is further amplified**. Empirically, GRPO+RLG **improves alignment** despite lacking strict $\beta/\omega$ equivalence.
>
> Following the reviewer's suggestion, we conducted an additional 1D case study (now included in Appendix E), using a Gaussian-mixture base distribution with reward function $r(x) = -0.1|x-4|$, and compared both standard policy-gradient RL and GRPO under identical conditions. For policy-gradient RL, the learned distribution closely matches the theoretical optimum $\pi^{*} \propto \pi_{\mathrm{ref}} \exp(r/\beta)$, and applying RLG generates **distributional shifts** that precisely follow the theoretically predicted transformation $\beta \rightarrow \beta/\omega$. In contrast, GRPO's learned distribution, as expected, deviates from the exponential-tilting form due to its fundamentally different group-advantage objective. Consequently, the $\beta/\omega$ equivalence does not hold for GRPO. However, our experiments show that RLG still **consistently shifts** the GRPO-trained policy toward **higher-reward regions** as $\omega$ increases. This demonstrates that, even when the closed-form theoretical justification does not strictly apply, RLG reliably acts as an **inference-time reward-extrapolation mechanism**. Thus, the 1D results not only validate our theoretical claims in the regimes where they apply, but also provide **direct empirical evidence** that RLG remains effective for non-KL-regularized algorithms such as GRPO.
>
> Finally, across all downstream experiments—including challenging tasks and non-idealized algorithms like GRPO—we observe **consistent practical gains** from RLG. These empirical results indicate that the theoretical limitations noted by the reviewer represent acknowledged constraints rather than fundamental flaws. While a gap exists between idealized analysis and real-world model behavior, RLG nonetheless provides **substantial practical value** as an effective, **training-free alignment-control mechanism**.
>
> In summary, while the theoretical analysis provides intuition rather than guarantees, the **extensive empirical evidence**—including both controlled and large-scale downstream experiments—clearly shows that RLG remains **effective and valuable in practice**, even when the ideal assumptions do not strictly hold.

---

> ### Author Response · Authors · 2025-11-21
> **Official Comment by Authors**
>
> **Q2**: “Indeed there are many studies … including several papers currently under review with this year’s ICLR … Those studies appear to use substantially more sophisticated techniques …”
>
> **A**: We want to **emphasize** that using other submissions currently under **double-blind review at the same conference** as a basis for negative evaluation is **explicitly prohibited by ICLR reviewing ethics**.
>
> *We consider papers contemporaneous if they are published within the last two months. That means, since our full paper deadline is September 24, if a paper was published (i.e., at a peer-reviewed venue) on or after July 24, 2025, **authors are not required to compare their own work to that paper**. Note that arXiv is **not considered a peer-reviewed venue**. As such, authors are not required to compare to papers solely on arXiv: they may be excused for not knowing about papers not published in peer-reviewed conference proceedings or journals, which includes papers exclusively available on arXiv.*
>
> Regarding legitimate concurrent work, we have **already addressed all publicly available papers**. Our original manuscript contains a dedicated **"Concurrent Work"** section in which we discuss CFGRL[4] and Diffusion Blend[5], both released shortly before our submission. Although authors are not required to include comparisons against newly released work on Arxiv, we **nonetheless provided detailed discussion of publicly accessible papers**, **clarified theoretical distinctions**, and demonstrated **substantially broader empirical coverage across architectures, RL algorithms, and alignment tasks**. During the OpenReview discussion phase, we further added extended results—such as **Blend-style interpolation**—to strengthen completeness and transparency.
>
> By contrast, comparisons against anonymous submissions under review cannot be meaningfully addressed. The reviewer's remark that "several ICLR submissions use more sophisticated techniques" is **impossible** for authors to evaluate: we cannot access these manuscripts, they are **unpublished and anonymous**, and they cannot be part of any fair or actionable comparison. Even if identifiers had been provided, this type of comparison would still violate double-blind reviewing norms.
>
> We therefore respectfully ask the reviewer to **reassess the evaluation based on public**, citable literature. Our work introduces a theoretically grounded unification of score interpolation and KL-controlled RL alignment, a simple and general inference-time control mechanism for diffusion and flow models, and extensive empirical results across diverse tasks. These contributions should be **judged relative to published and accessible work**, as required by ICLR's reviewing guidelines, rather than against anonymous submissions that authors cannot possibly engage with.
>
> [1]Guiding a Diffusion Model with a Bad Version of Itself
>
> [2]CFG-Zero*: Improved Classifier-Free Guidance for Flow Matching Models
>
> [3]CFG++: Manifold-constrained Classifier Free Guidance for Diffusion Models
>
> [4]Diffusion Guidance Is a Controllable Policy Improvement Operator
>
> [5]Diffusion Blend: Inference-Time Multi-Preference Alignment for Diffusion Models

---

### Official Review · Reviewer_9UbK · 2025-10-31

**Soundness:** 2
**Presentation:** 2
**Contribution:** 2
**Rating:** 2
**Confidence:** 3

**Summary:**

This paper proposes Reinforcement Learning Guidance (RLG), an inference‑time control method for denoising‑based generators (diffusion and flow‑matching). Given a base model and an RL‑aligned model trained toward a downstream objective, RLG linearly combines their scores/velocities during sampling, with a scalar weight $w$ (Alg. 1). The paper argues theoretically that sampling with weight $w$ is equivalent to sampling from a distribution proportional to $p_{\text{ref}}^{\,1-w} p_{\theta}^{\,w}$, and—under the usual KL‑regularized RL objective—this corresponds to reducing the effective KL coefficient from $\beta$ to $\beta/w$. Empirically, RLG is claimed to (i) improve preference metrics (Aesthetic, ImageReward, PickScore) on SD1.5/DPO, SDXL/SPO and SD3.5‑M/GRPO; (ii) raise compositional accuracy on GenEval; (iii) increase OCR accuracy for text rendering; (iv) provide a tunable “slider” for image compressibility; and (v) help inpainting and personalization, all without extra training.

**Strengths:**

1. **Simple, training‑free control**: The method only needs the base and RL‑tuned models and a scalar $w$. The implementation is trivial (one extra forward pass and a weighted sum), yet it gives a convenient **post‑hoc** control over alignment strength.
2. **Unified score/velocity perspective**: The paper writes RLG both on **scores** and **flow velocities** and shows their equivalence via the SDE/ODE unification. This makes the method broadly applicable to DDPM/DDIM‑style diffusion and Flow Matching families.

**Weaknesses:**

- The core step $\hat s = (1-w)s_{\text{ref}} + w s_{\theta} = \nabla \log\big(p_{\text{ref}}^{1-w} p_{\theta}^{w}\big)$ is standard, but it does not imply the sampler actually draws from that target density under common discretizations; recent analyses showed CFG is a predictor‑corrector heuristic and typically does not sample $p(x|c)^{\gamma} p(x)^{1-\gamma}$. This limitation carries over to RLG, especially when $w>1$ (negative exponent on $p_{\text{ref}}$) [1][2].
- Equivalence to $\beta/w$ depends on the RL objective: The $\beta \mapsto \beta/w$ argument assumes the optimal KL‑regularized policy $p_{\theta} \propto p_{\text{ref}} \exp(R/\beta)$. The paper evaluates on GRPO and DPO models as well, where this assumption may not hold exactly; thus the “effective‑$\beta$” interpretation is at best approximate for these cases [3].
- Tables report mean scores and “win‑rates” but no confidence intervals/variance estimates or significance tests. Several gains are modest (e.g., ImageReward often plateaus or slightly drops at large $w$), so the practical significance is unclear without error bars and multiple seeds.
- Interplay with CFG left largely unexplored: RLG is applied on top of a fixed CFG scale but the coupling between CFG and RLG can be non‑trivial; without ablations over (CFG, $w$) grids, it is hard to know when each dominates or conflicts. [4]

[1] Feynman-Kac Correctors in Diffusion: Annealing, Guidance, and Product of Experts, https://arxiv.org/abs/2503.02819

[2] Classifier-Free Guidance is a Predictor-Corrector, https://arxiv.org/abs/2408.09000

[3] Direct Preference Optimization: Your Language Model is Secretly a Reward Model, https://arxiv.org/abs/2305.18290

[4] Guided Flows for Generative Modeling and Decision Making, https://arxiv.org/abs/2311.13443

**Questions:**

1. Can you clarify whether your sampler with RLG actually converges to $p_{\text{ref}}^{1-w} p_{\theta}^{w}$? Have you tried Feynman–Kac correctors or other product‑of‑experts correctors to make the interpolation distributionally sound?
2. When $\beta/w$ holds: For DPO and GRPO‑tuned models, what is the theoretically correct “effective $\beta$” story, if any? Can you include a small controlled experiment showing where the $\beta/w$ prediction fails (or holds approximately)?
3. Could RLG extend to multi‑objective blending (e.g., combine two RL‑aligned experts with the base at inference)?
4. How do RLG's latency and energy consumption compare to single-model sampling? RLG doubles forward passes per step (base + RL policy). The paper fixes 20 steps, which masks the runtime cost relative to the RL‑only baseline.

---

> ### Author Response · Authors · 2025-11-21
> **Official Comment by Authors**
>
> **Q1**: CFG is a predictor-corrector heuristic. It will not sample from the targeted distribution. Please clarify whether your sampler with RLG actually converges. Have you tried other product‑of‑experts correctors to make the interpolation distributionally sound?
>
> **A**: We appreciate the reviewer's theoretical perspective on CFG and RLG.
>
> In response to the reviewer's suggestion, we did experiment with a more distributionally principled sampler inspired by the predictor–corrector view of CFG, specifically the **PCG sampler** introduced in *CFG is a Predictor–Corrector*[1]. Concretely, starting from our standard RLG update at each diffusion step, we add an extra **Langevin-based corrector step** in the direction of the RL expert. We use 0.001 as Langevin-based corrector step size.
>
> This PCG-style sampler is **more expensive**: it requires an extra gradient evaluation (and hence an additional forward–backward pass) per step on top of the usual base+RL evaluations, further increasing the already higher cost of RLG. We therefore evaluated whether it provides a tangible benefit over our simpler RLG sampler on the DrawBench pickscore GRPO setting. Table 1 compares Aesthetic and PickScore for several RLG weights, with and without the PCG corrector.
>
> | $w_{\text{RL}}$ | Aesthetic (PCG) | Aesthetic (RLG) | PickScore (PCG) | PickScore (RLG) |
> |:---------------:|:---------------:|:---------------:|:---------------:|:---------------:|
> | 1.0             | 6.394           | 6.45            | 23.282          | 23.29           |
> | 1.6             | 6.397           | 6.57            | 23.289          | 23.53           |
> | 2.0             | 6.399           | 6.62            | 23.293          | 23.57           |
> | 2.4             | 6.402           | 6.66            | 23.297          | 23.58           |
> | 3.0             | 6.405           | 6.68            | 23.305          | 23.56           |
>
> *Table 1: Comparison of our standard RLG sampler vs. a PCG-style predictor–corrector sampler.*
>
> As the table shows, the PCG-based corrector produces metrics that are **essentially indistinguishable** from those of our standard RLG sampler, while incurring a non-trivial additional computational cost and implementation complexity. For these reasons, we chose **not** to adopt the PCG/Langevin corrector as our default sampler, and instead to keep RLG as a simple, test-time guidance knob in the spirit of CFG.

---

> ### Author Response · Authors · 2025-11-21
>
> **Q2**: $\beta$ converging to $\beta / \omega$ assumption will not hold for **GRPO and DPO**. Can you include a small controlled experiment showing where the beta/w prediction fails (or holds approximately)?
>
> **A**: We thank the reviewer for raising the important question regarding the universal applicability of the $\beta \rightarrow \beta/\omega$ assumption for GRPO and DPO. We provide the following clarifications:
>
> **DPO follows the theoretical optimal RL solution.**
>
> As established in the **original DPO** derivation [2] (specifically Equation (4) and Appendix A.1), for RL algorithms with the KL-constrained reward maximization objective:
>
> $$
> \max_{\pi_\theta} \; E_{x \sim D, y \sim \pi_\theta} \left[ r_\phi(x, y) \right] - \beta \, KL \left[ \pi_\theta(y \mid x) \,\|\, \pi_{ref}(y \mid x) \right].
> $$
>
> The theoretical optimum of this objective takes the form:
>
> $$
> \pi_r(y\mid x) = \frac{1}{Z(x)}\pi_{ref}(y\mid x)\exp\left(\frac{1}{\beta}r(x, y)\right),
> $$
>
> This confirms that **DPO provably converges to this exponential-tilting solution**. Therefore, adjusting the RLG scale $\omega$ is **mathematically equivalent to modifying the effective KL coefficient as $\beta/\omega$**, making our theoretical analysis fully **applicable**.
>
> **GRPO does not follow the exponential-tilting form but maintains reward-seeking properties.**
>
> Due to GRPO's group-advantage formulation, its optimal policy does not satisfy the exponential-tilting relationship $\pi^{*} \propto \pi_{\mathrm{ref}} \exp(r/\beta)$. Recent theoretical analysis confirms that GRPO follows different optimality conditions [3]. Nevertheless, GRPO consistently shifts probability mass toward higher-reward regions. When combined with RLG—which performs inference-time reward extrapolation—this reward-seeking behavior is further amplified. **Empirically, GRPO+RLG demonstrates improved alignment** despite the lack of strict $\beta/\omega$ equivalence.
>
> **Controlled 1D experiment validates our theoretical claims.**
>
> Following the reviewer's suggestion, we conducted **a one-dimensional case study** (now included in Appendix E) using a Gaussian-mixture base distribution with reward function $r(x) = -0.1 |x-4|$, comparing policy-gradient RL and GRPO baselines.
>
> For policy gradient RL, the learned distribution closely matches the theoretical optimum $\pi^{*} \propto \pi_{\mathrm{ref}} \exp(r/\beta)$, and **RLG produces distribution shifts that precisely match the $\beta\to\beta/\omega$ transformation prediction**. In contrast, GRPO's learned distribution deviates from the exponential-tilting solution, consistent with its non-standard objective. Consequently, the $\beta/\omega$ prediction doesn't strictly hold for GRPO. However, we observe that RLG still **consistently shifts the GRPO policy toward higher-reward regions as $\omega$ increases**, confirming RLG's effectiveness as an inference-time reward extrapolation mechanism even without closed-form theoretical support.
>
> These additional experiments and discussion now clearly distinguish between cases where the theoretical equivalence $\beta \rightarrow \beta/\omega$ holds (DPO, policy gradient-style RL) and cases where it doesn't (GRPO), while empirically demonstrating RLG's consistent reward-extrapolation behavior across different algorithms.

---

> ### Author Response · Authors · 2025-11-21
> **Official Comment by Authors**
>
> **Q3**: practical significance is unclear without error bars and multiple seeds
>
> **A**: We thank the reviewer for pointing out the need for error bars and multi-seed evaluation to assess practical significance. In the revision, we add multi-seed experiments and report mean ± standard error of the mean (SEM) for the key preference-alignment metrics.
>
> Concretely, we evaluate on the full DrawBench[4] prompt test set using five random seeds for two representative models: (i) the *Flow-GRPO*[5] model, which exhibits the largest gains from RLG, and (ii) the *SPO*[6] model, which exhibits the smallest gains. These two settings therefore **bracket the range of observed behaviors**. For each model, we compare the RL-only configuration ($w=1.0$) against the best-performing RLG configuration.
>
> Table 1 reports the multi-seed means and SEMs. For Flow-GRPO, RLG with $w=2.2$ improves both Aesthetic and PickScore over the RL-only model ($w=1.0$), while ImageReward remains essentially unchanged. For SPO, RLG again improves Aesthetic and ImageReward; the changes in PickScore **are smaller in magnitude**. In all cases, the SEMs are **two to three orders of magnitude smaller than the absolute differences between configurations**, indicating that the gains are **robust to random seed variation** rather than being artifacts of a single run.
>
> | Model | $w$ | Aesthetic $\uparrow$ | PickScore $\uparrow$ | ImageReward $\uparrow$ |
> |:------|:----|:---------------------|:---------------------|:-----------------------|
> | Flow-GRPO (RL-only) | 1.0 | $6.44 \pm 0.0022$ | $23.28 \pm 0.0040$ | $1.40 \pm 0.0017$ |
> | Flow-GRPO + RLG | 2.2 | $6.64 \pm 0.0015$ | $23.58 \pm 0.0029$ | $1.39 \pm 0.0030$ |
> | SPO (RL-only) | 1.0 | $6.42 \pm 0.0012$ | $23.69 \pm 0.0069$ | $1.13 \pm 0.0046$ |
> | SPO + RLG | 1.4 | $6.49 \pm 0.0026$ | $22.73 \pm 0.0054$ | $1.15 \pm 0.0037$ |
>
> *Table 1: Multi-seed results on the DrawBench aesthetics alignment task. We report mean ± SEM over five random seeds for Flow-GRPO and SPO, with and without RLG (RLG weight $w$).*
>
> To further quantify statistical significance, we perform **paired one-sided $t$-tests over seeds**, comparing RLG against the RL-only baseline for each metric and model. The resulting $t$-statistics and one-sided **$p$-values** are summarized in Table 2. For Flow-GRPO, RLG yields highly significant improvements in Aesthetic and PickScore, while the difference in ImageReward is not significant, consistent with the nearly identical means. For SPO, RLG is significantly better than the RL-only baseline on all three metrics. These results demonstrate that RLG provides statistically and practically significant gains over RL-only fine-tuning, even when accounting for variability across multiple seeds.
>
> | Model | Metric | $t$ | $p_{\text{one-sided}}$ |
> |:------|:-------|:----|:-----------------------|
> | Flow-GRPO | Aesthetic | 266.17 | $5.98 \times 10^{-10}$ |
> | Flow-GRPO | PickScore | 93.85 | $3.86 \times 10^{-8}$ |
> | Flow-GRPO | ImageReward | -4.04 | $9.92 \times 10^{-1}$ |
> | SPO | Aesthetic | 24.49 | $8.25 \times 10^{-6}$ |
> | SPO | PickScore | 9.87 | $2.95 \times 10^{-4}$ |
> | SPO | ImageReward | 5.27 | $3.12 \times 10^{-3}$ |
>
> *Table 2: Paired one-sided $t$-tests over five random seeds comparing RLG vs. RL-only for each metric on the DrawBench aesthetics task. We report the $t$-statistic and one-sided $p$-value $p_{\text{one-sided}}$.*

---

> ### Author Response · Authors · 2025-11-21
> **Official Comment by Authors**
>
> **Q4**: RLG is applied on top of a fixed CFG scale but the coupling between CFG and RLG can be non‑trivial; without ablations over (CFG, RLG) grids, it is hard to know when each dominates or conflicts.
>
> **A**: We agree that the coupling between classifier-free guidance (CFG) and RLG could be non-trivial, and that joint ablations over (CFG, RLG) grids are needed to understand potential dominance or conflicts. Accordingly, we conduct a systematic grid search on the DrawBench aesthetics GRPO task, varying both CFG scale and RLG scale.
>
> We **sweep CFG scale ∈ {2.0, 3.0, 4.0, 4.5, 5.0} and RLG scale ∈ {1.4, 1.8, 2.2, 2.4, 3.0}**, evaluating PickScore on the full DrawBench test set. The results (Table 1) reveal clear trends: for any **fixed CFG scale**, increasing RLG scale from 1.4 to 2.2–2.4 **consistently improves PickScore before saturating**, showing that RLG robustly enhances performance without conflicting with CFG. Similarly, for fixed RLG scale, intermediate CFG values (4.0–4.5) outperform extremes. Our default configuration (CFG=4.5, RLG=2.4) lies on the **global optimum plateau**, achieving top PickScore (23.59), demonstrating that CFG and RLG can be tuned to complementary, maximized effect.
>
> | CFG \ RLG $w$ | 1.4   | 1.8   | 2.2   | 2.4     | 3.0   |
> |:-------------:|:-----:|:-----:|:-----:|:-------:|:-----:|
> | 2.0           | 23.14 | 23.27 | 23.31 | 23.32   | 23.32 |
> | 3.0           | 23.42 | 23.50 | 23.54 | 23.55   | 23.54 |
> | 4.0           | 23.48 | 23.56 | 23.58 | **23.59** | 23.57 |
> | 4.5 (default) | 23.48 | 23.56 | 23.57 | **23.59** | 23.58 |
> | 5.0           | 23.47 | 23.55 | 23.57 | 23.58   | 23.57 |
>
> *Table 1: PickScore on the DrawBench aesthetics GRPO task for a grid of CFG and RLG scales.*
>
> Moreover, when we compare RLG-enhanced models to the corresponding RL-only baselines at fixed CFG scales, we observe that enabling RLG consistently improves both aesthetic metrics (Aesthetic score and PickScore) across all tested CFG values. In other words, RLG does not merely compensate for a suboptimal CFG choice; it provides an additional, consistent boost even when the underlying CFG scale is already well tuned.
>
> To more formally analyze whether CFG and RLG *independently* contribute to performance, we conduct a **two-way ANOVA** with factors *rlg_scale* and *cfg_scale* on both *aesthetic_score* and *pickscore_score*. As shown in Table 2, for *aesthetic_score* we find a **very large main effect of RLG** ($F = 2960.56$, $p = 9.79 \times 10^{-28}$) and also a significant, though smaller, main effect of CFG ($F = 280.79$, $p = 2.75 \times 10^{-19}$). For *pickscore_score*, both factors again show **strong main effects**: RLG ($F = 134.98$, $p = 6.97 \times 10^{-14}$) and CFG ($F = 206.26$, $p = 6.69 \times 10^{-18}$). The $F$-statistic measures how much variance is explained by a factor relative to the residual variance; all $F$ values are well above $10^2$, indicating that both CFG and RLG consistently and robustly influence performance. Across metrics, the main effects of RLG and CFG are of the same order of magnitude (e.g., for Aesthetic: $2960$ vs. $280$; for PickScore: $135$ vs. $206$), showing that neither factor is negligible when both are jointly tuned.
>
> | Metric | Factor | $F$ | $p$ |
> |:-------|:-------|:----|:----|
> | Aesthetic score | RLG scale | 2960.56 | $9.79 \times 10^{-28}$ |
> | Aesthetic score | CFG scale | 280.79 | $2.75 \times 10^{-19}$ |
> | PickScore score | RLG scale | 134.98 | $6.97 \times 10^{-14}$ |
> | PickScore score | CFG scale | 206.26 | $6.69 \times 10^{-18}$ |
>
> *Table 2: Two-way ANOVA main effects of RLG scale and CFG scale on Aesthetic and PickScore metrics for the DrawBench aesthetics GRPO task.*
>
> Finally, to test whether RLG and CFG **interact** in a systematic way, or whether they primarily act as additive main effects, we fit for each metric a linear regression model with the following form:
>
> **metric = β₀ + β₁ × rlg_scale + β₂ × cfg_scale + β₃ × (rlg_scale × cfg_scale) + ε**
>
> where the response is either *aesthetic_score* or *pickscore_score*, and the predictors are the numeric RLG scale, CFG scale, and their interaction term. For *aesthetic_score*, the regression achieves an excellent fit **($R^2 = 0.984$, overall $F = 588.8$, $p = 2.18 \times 10^{-25}$)**, and the **interaction coefficient $\beta_3$ is small and not statistically significant** ($t = 1.44$, $p = 0.161$). For *pickscore_score*, the model explains substantial variance ($R^2 = 0.638$, $F = 16.47$, $p = 2.31 \times 10^{-6}$), and again the **interaction term is non-significant ($t = -0.82$, $p = 0.418$)**. These results indicate that RLG and CFG largely contribute as **additive main effects**, with no detectable systematic interaction. This directly addresses the reviewer's concern: we observe no regimes where RLG and CFG systematically conflict; both can be jointly tuned to reach a shared optimum, providing strong, statistically significant contributions to performance.

---

> ### Author Response · Authors · 2025-11-21
> **Official Comment by Authors**
>
> **Q5**: Could RLG extend to multi‑objective blending (e.g., combine two RL‑aligned experts with the base at inference)?
>
> **A**: We appreciate the reviewer's suggestion to investigate whether RLG can extend to multi-objective blending by combining multiple RL-aligned experts at inference. In the revision, we add experiments that **fuse an aesthetics-aligned RLG expert and an OCR-aligned RLG expert,** and evaluate on a held-out subset of the OCR test prompt set.
>
> Concretely, we consider two RLG experts trained with different reward functions: (i) an *Aesthetic* expert aligned to PickScore-style preference metrics, and (ii) an *OCR* expert aligned to OCR quality. At inference, we combine their guidance on top of the same base model via a linear blend with non-negative weights $w_1$ (aesthetic expert) and $w_2$ (OCR expert). Table 1 reports results for several representative configurations.
>
> The single-objective endpoints illustrate the **inherent trade-off**: when we use only the aesthetics expert $(w_1=1.0, w_2=0)$, we obtain strong aesthetic quality (5.76 Aesthetic, 23.76 PickScore) but relatively low OCR performance (0.592). Once we enable multi-objective fusion $(w_1>0, w_2>0)$, the "other" objective improves markedly compared to its single-expert baseline.
>
> | Setting | $w_1$ | $w_2$ | Aesthetic $\uparrow$ | PickScore $\uparrow$ | OCR $\uparrow$ |
> |:--------|:------|:------|:---------------------|:---------------------|:---------------|
> | Aesthetics-only | 1.0 | 0.0 | 5.76 | 23.76 | 0.592 |
> | OCR-only | 0.0 | 1.0 | 5.11 | 22.24 | 0.936 |
> | Fuse (balanced) | 1.0 | 1.0 | 5.51 | 23.44 | 0.926 |
> | Fuse (OCR-heavy) | 1.0 | 1.8 | 5.38 | 22.88 | 0.940 |
> | Fuse (Aes-heavy) | 2.4 | 1.0 | 5.76 | 23.99 | 0.894 |
>
> *Table 1: Multi-objective RLG fusion of aesthetics and OCR experts on a subset of the OCR test prompt set. $w_1$ and $w_2$ control the relative strength of the aesthetics-aligned and OCR-aligned experts*
>
> We also investigate an extrapolation regime where **we fix the OCR expert weight and increase the aesthetics expert weight $w_1$**. Table 2 shows the effect of sweeping $w_1$ at fixed $w_2=1.0$. As $w_1$ increases from 0 to 2.4, Aesthetic and PickScore improve monotonically (from 5.11 to 5.76 in Aesthetic and from 22.24 to 23.99 in PickScore), while OCR performance stays in a high regime (0.936 down to 0.894), still far above the aesthetics-only baseline at $w_2=0$ (0.592). In other words, by leveraging multi-expert RLG fusion we can "dial up" aesthetics quality via $w_1$ **without collapsing OCR performance** back to the level of an aesthetics-only model.
>
> | $w_1$ | $w_2$ | Aesthetic $\uparrow$ | PickScore $\uparrow$ | OCR $\uparrow$ |
> |:------|:------|:---------------------|:---------------------|:---------------|
> | 0.0   | 1.0   | 5.11 | 22.24 | 0.936 |
> | 1.0   | 1.0   | 5.51 | 23.44 | 0.926 |
> | 1.8   | 1.0   | 5.68 | 23.86 | 0.894 |
> | 2.4   | 1.0   | 5.76 | 23.99 | 0.894 |
>
> *Table 2: Effect of increasing the aesthetics expert weight $w_1$ while fixing the OCR expert weight $w_2=1.0$ on a subset of the OCR test prompt set. As $w_1$ increases, Aesthetic and PickScore steadily improve, while OCR remains high and significantly above the aesthetics-only baseline (OCR = 0.592 at $w_2=0$).*
>
> Overall, these results demonstrate that RLG naturally **extends to multi-objective blending**: by combining multiple RL-aligned experts with tunable fusion weights $(w_1, w_2)$ on top of a shared base model, we can trade off between objectives and obtain configurations that simultaneously perform well on both aesthetics and OCR, improving the weaker objective relative to single-expert baselines while preserving most of the strength of the stronger one.

---

> ### Author Response · Authors · 2025-11-21
> **Official Comment by Authors**
>
> **Q6**: How do RLG's latency and energy consumption compare to single-model sampling? RLG doubles forward passes per step (base + RL policy). The paper fixes 20 steps, which masks the runtime cost relative to the RL‑only baseline.
>
> **A**: We thank the reviewer for raising the question of runtime and energy overhead. In the main paper we fixed the number of diffusion steps (20) across all methods in order to isolate quality differences; here we explicitly quantify latency and compare RLG against single-model sampling and naive test-time scaling.
>
> On a single H100 GPU (same batch size and resolution as in the main experiments), sampling with our 20-step diffusion base model takes about **9 seconds per batch**, whereas RLG with $w=2.2$ and the same 20 denoising steps takes about **17 seconds**. This **~1.9× latency increase** is expected, since RLG performs two forward passes per step (base model + RL-aligned policy), effectively doubling the number of neural evaluations.
>
> Importantly, RLG acts as a simple **test-time scaling** knob: by enabling RLG at inference (and choosing the weight $w$), users can **trade more compute for higher alignment quality**, without retraining or changing the base sampler. To understand whether the same quality gains could be achieved by simply increasing the diffusion step count for a single RL-only model, we compare GRPO on the DrawBench aesthetics task under different test-time budgets. Table 1 reports Aesthetic and PickScore for (i) the RL-only model at 20/40/60 steps and (ii) RLG with $w=2.2$ at 20 steps.
>
> Two observations emerge. First, **naive test-time scaling by increasing the number of denoising steps from 20 to 40 or 60 has essentially no effect on performance**: the RL-only model's Aesthetic and PickScore remain flat or even slightly decrease when going from 20 to 60 steps. Second, RLG at 20 steps (which incurs roughly the same number of forward passes as a 40-step RL-only model, since it uses two networks per step) yields a **clear improvement**: Aesthetic increases from 6.45 (RL 20 steps) to 6.64, and PickScore from 23.29 to 23.58. RLG at 20 steps also outperforms RL-only at 40 and 60 steps, despite having a comparable or smaller effective compute budget.
>
> | Method | Diffusion steps | Effective forward passes | Aesthetic $\uparrow$ | PickScore $\uparrow$ |
> |:-------|:---------------:|:------------------------:|:--------------------:|:--------------------:|
> | RL-only | 20 | 20 | 6.45 | 23.29 |
> | RL-only | 40 | 40 | 6.41 | 23.30 |
> | RL-only | 60 | 60 | 6.40 | 23.29 |
> | RLG ($w=2.2$) | 20 | 40 | 6.64 | 23.58 |
>
> *Table 1: Quality vs. test-time budget comparison. RLG at 20 steps (40 effective forward passes) significantly outperforms RL-only models even at higher step counts, demonstrating that the additional compute is used effectively.*
>
> In summary, while RLG does introduce an expected ~2× cost in latency and energy relative to a single 20-step model, this additional test-time compute is substantially more effective than simply adding more diffusion steps: with roughly the same number of forward passes as a 40-step sampler, **RLG delivers a significant gain in preference alignment**.
>
> [1]Classifier-Free Guidance is a Predictor-Corrector
>
> [2]Direct Preference Optimization: Your Language Model is Secretly a Reward Model
>
> [3]What is the Alignment Objective of GRPO?
>
> [4]Photorealistic Text-to-Image Diffusion Models with Deep Language Understanding
>
> [5]Flow-GRPO: Training Flow Matching Models via Online RL
>
> [6]Aesthetic Post-Training Diffusion Models from Generic Preferences with Step-by-step Preference Optimization

---

> > ### Comment · Reviewer_9UbK · 2025-11-24
> >
> > I appreciate the authors' efforts; it's very impressive to see the new experimental results demonstrating the effectiveness of the method. Most of my concerns have been addressed. However, the authors seem to have omitted any explanation or proof regarding whether the sampler with RLG converges to $p_{\text{ref}}^{1-w} p_{\theta}^{w}$. Based on these points, I will increase my rating.

---

### Official Review · Reviewer_oiuS · 2025-11-01

**Soundness:** 3
**Presentation:** 3
**Contribution:** 3
**Rating:** 4
**Confidence:** 4

**Summary:**

This paper reinterprets RL fine-tuning for diffusion models through the lens of stochastic differential equations and guidance/reward conditioning. The authors propose Reinforcement Learning Guidance (RLG): at inference time, one uses both the base diffusion model and an RL-fine-tuned version of it; then one combines their outputs via a geometric average. It has been shown that adjusting the guidance scale in RLG corresponds mathematically to adjusting a KL-regularization coefficient in the RL objective. Experiments have been conducted on SD3.5 and SDXL.

**Strengths:**

The paper is well-written, and I mostly enjoyed reading it. The main contribution/strength is to provide flexibility of inference-time control in fine-tuning. The numerical experiments also support the proposed approach.

**Weaknesses:**

One problem is the sensitivity of the weight $w_{RL}$: it seems (Table 1) that larger the weight is, the better the performance is. However,  large $w_{RL}$ may lead to distributional shift, and it is not clear to me how this can be interpreted, or how to choose the hyperparameter.

**Questions:**

See weaknesses.

---

> ### Author Response · Authors · 2025-11-21
> **Official Comment by Authors**
>
> **A**: We thank the reviewer for pointing out the apparent monotonic trend in Table 1 and for raising the concern about distributional shift at large RLG weights. We agree that our original table could be misread as suggesting that "larger is always better." In the revision, we extend the sweep over the RLG scale $w_{\text{RL}}$ and explicitly show where performance saturates and begins to **decrease**.
>
> Table 1 reports human-preference metrics for the SD3.5-M model series across a wider range of $w_{\text{RL}}$[1]. As before, $w_{\text{RL}}=0$ denotes the original base model, and $w_{\text{RL}}=1.0$ corresponds to the GRPO-finetuned policy[2]. As $w_{\text{RL}}$ increases from $0$ to around $2.4$–$2.6$, Aesthetic Score and PickScore steadily improve. Beyond this range, the gains saturate and we begin to see **small declines**: for example, PickScore slightly drops from $23.59$ to $23.56$.
>
> | **$w_{\text{RL}}$** | **Aesthetic Score ($\uparrow$)** | **ImageReward ($\uparrow$)** | **PickScore ($\uparrow$)** |
> |:-------------------:|:--------------------------------:|:----------------------------:|:--------------------------:|
> | 1.2                 | 6.48                             | 1.41                         | 23.36                      |
> | 1.4                 | 6.54                             | 1.40                         | 23.48                      |
> | 1.6                 | 6.57                             | 1.40                         | 23.53                      |
> | 1.8                 | 6.60                             | 1.41                         | 23.56                      |
> | 2.0                 | 6.62                             | 1.40                         | 23.57                      |
> | 2.2                 | 6.64                             | 1.39                         | 23.58                      |
> | 2.4                 | 6.66                             | 1.39                         | 23.58                      |
> | 2.6                 | 6.68                             | 1.37                         | 23.59                      |
> | 2.8                 | 6.68                             | 1.36                         | 23.56                      |
>
> *Table 1: Mean scores for the SD3.5-M model series on human preference metrics across various RLG scales ($w_{\text{RL}}$). The scale $w_{\text{RL}}=0.0$ corresponds to the original SD3.5-M base model, while $w_{\text{RL}}=1.0$ represents the model after GRPO finetuning. The best PickScore is obtained around $w_{\text{RL}}=2.6$, after which performance starts to slightly decrease.*
>
> We interpret this behavior as follows. In practice, the RL expert is trained with finite data and an **approximate on-policy algorithm (GRPO) under a particular KL regularization**; the resulting policy $\pi_{\text{RL}}$ is only an approximation to the ideal reward-weighted distribution. For very large $w_{\text{RL}}$, we effectively *over-trust* this imperfect approximation: any biases or artifacts in $\pi_{\text{RL}}$ (e.g., overuse of certain layouts, textures, or compositions that the reward happens to like) are amplified, and the samples start to drift away from the base distribution, which manifests as saturation and eventual decline in Aesthetic and PickScore.
>
> A second source of approximation is the guidance form itself. Our inference rule combines the base and RL experts in a **CFG-style way**. Just as classical CFG is known *not* to be an exact sampler for the true conditional distribution and can distort samples when the scale is very high[3], our RLG guidance is also an approximate mechanism: it is designed to be **simple and effective at test time**. Consequently, very large $w_{\text{RL}}$ can introduce distributional shift and degrade sample quality, exactly as extremely **large CFG scales can hurt realism**.
>
> Because of these limitations, we treat it as a test-time knob analogous to CFG. In practice, we select $w_{\text{RL}}$ via a simple hyperparameter search on a small validation prompt set, using the same reward that was used for RL training. This procedure is described in detail in our answer to Q2: we can grid-search several candidate weights and the chosen value consistently matches the optimum found on the full evaluation set. This tuning step keeps us away from pathological large weights and effectively mitigates the distribution-shift issues discussed above.

---

> ### Author Response · Authors · 2025-11-21
> **Official Comment by Authors**
>
> **Q2**. How should the weight hyperparameter be selected? Can the authors provide clearer guidance or justification for choosing it?
>
> **A**: A key advantage of RLG is that the same reward used for RL training is available at test time, so $w_{\text{RL}}$ can be **selected by simple validation on a fixed prompt set**, without additional human labeling.
>
> Our recommended procedure is:
>
> 1.  **Fix a small validation set** of prompts (e.g., 32–64 prompts) drawn from the same distribution as the intended use case. Because the reward is automatically computed, this set can be much smaller than a typical test set.
> 2.  **Grid search over candidate weights** (e.g., $w_{\text{RL}} \in \{0.0, 0.8, 1.2, \ldots, 2.8\}$). For each candidate weight, generate images with RLG and compute the relevant reward(s) and auxiliary metrics (Aesthetic, PickScore, etc.) on this validation set.
> 3.  **Select the optimal value** of $w_{\text{RL}}$ that maximizes the reward or yields the best trade-off between metrics, and then keep this value fixed for all subsequent evaluations.
>
> Because the **reward is available at inference**, this grid search is very **cheap**. For example, on a single H100 GPU, generating images for a 32-prompt validation set at a given $w_{\text{RL}}$ takes well under one minute; sweeping over a handful of candidate weights is therefore practical even in production settings. In our experiments, tuning on a small subset (e.g., 32 prompts) consistently identifies the same optimal range as tuning on the full test set.
>
> In summary, rather than being a brittle parameter, $w_{\text{RL}}$ can be treated **exactly like a CFG scale**: it is a test-time knob that can be robustly selected via **a simple grid search on a small, automatically scored validation set**, and it exhibits a broad optimum region where performance is stable.
>
> [1]Introducing Stable Diffusion 3.5
>
> [2]Flow-grpo: Training flow matching models via online rl
>
> [3]Classifier-Free Guidance is a Predictor-Corrector

---

### Meta-Review · Area_Chair_NNRF · 2026-01-08

**Summary:**

Reviewers raised concerns on theoretical gaps (e.g., sampler convergence, optimal policy assumptions, GRPO applicability), novelty as a CFG extension, w_RL sensitivity/selection, lack of error bars/significance, compute costs, CFG interplay, multi-objective extensions, and concurrent work comparisons. The rebuttal addressed practical issues like w_RL tuning, added experiments (e.g., PCG convergence, multi-seed stats, CFG grids, 1D GRPO studies, multi-objective fusion), and clarified ethics on citations. However, core theoretical limitations (e.g., no convergence proof, unrealistic optimality) and modest novelty remain unresolved.

**Reviewer Concerns:**

The rebuttal resolved concerns on w_RL sensitivity/selection with extended sweeps showing saturation, practical grid-search procedures, and CFG-like interpretations of shifts; added convergence experiments (e.g., PCG sampler comparisons with negligible gains); included error bars, multi-seed statistics, and significance tests demonstrating robust improvements; quantified runtime/energy overheads (~1.9x latency but efficient quality gains); conducted CFG-RLG grid ablations confirming additive benefits; extended to multi-objective blending with new fusion results; justified RLG for GRPO via 1D controlled studies; addressed reward hacking with extrapolations showing quality declines at extreme scales; fixed formatting/notations. On concurrent work, authors discussed published papers (e.g., CFGRL, Diffusion Blend) and noted ethical constraints against citing under-review submissions.

However, novelty and contribution debates persist, as RLG is perceived as a simple CFG variant with limited theoretical sophistication compared to advanced (though sometimes uncitable) alignment methods; the optimal policy assumption remains unrealistic, with empirical mitigations not fully closing the gap; no formal proof of sampler convergence or deeper analysis of extreme extrapolation limits and RL under-optimization.

**Reviewer Scores:**

(1) Reviewer oiuS: Initial rating 4. The rebuttal addressed the hyperparameter / saturation issue and improved experimental clarity. I expect this reviewer would raise to 5.
(2) Reviewer 9UbK: Initial rating 2. The reviewer posted that most concerns were addressed and that they would increase their rating (no exact final number posted). Given remaining question about formal convergence but good empirical fixes, I estimate 4.
(3) Reviewer gfqT: Initial rating 2 and remaining skeptical about theoretical depth/novelty. The rebuttal clarified empirical behavior but did not fully resolve theoretical limits. I estimate a modest bump to 3.
(4) Reviewer 2tey: Initial rating 4, explicit comment that they raise their score to 6 after the rebuttal. I take this as confirmed.

---

### Decision · Program_Chairs · 2026-01-26

Reject